# Inducing multiple nicks promotes interhomolog homologous recombination to correct heterozygous mutations in somatic cells

Akiko Tomita [1], Hiroyuki Sasanuma[2], Tomoo Owa [1], Yuka Nakazawa [3,4], Mayuko Shimada[3,4], Takahiro Fukuoka[3,5], Tomoo Ogi [3,4] & Shinichiro Nakada [1,6] ✉

CRISPR/Cas9-mediated gene editing has great potential utility for treating genetic diseases. However, its therapeutic applications are limited by unintended genomic alterations arising from DNA double-strand breaks and random integration of exogenous DNA. In this study, we propose NICER, a method for correcting heterozygous mutations that employs multiple nicks (MNs) induced by Cas9 nickase and a homologous chromosome as an endogenous repair template. Although a single nick near the mutation site rarely leads to successful gene correction, additional nicks on homologous chromosomes strongly enhance gene correction efficiency via interhomolog homologous recombination (IH-HR). This process partially depends on BRCA1 and BRCA2, suggesting the existence of several distinct pathways for MN-induced IH-HR. According to a genomic analysis, NICER rarely induces unintended genomic alterations. Furthermore, NICER restores the expression of disease-causing genes in cells derived from genetic diseases with compound heterozygous mutations. Overall, NICER provides a precise strategy for gene correction.

Genome editing enables targeted genome modification by inducing intentional DNA damage in specific genomic regions and by activating DNA repair mechanisms. CRISPR/Cas9 is a versatile gene editing tool. The *Streptococcus pyogenes* Cas9 (Cas9)–guide RNA (gRNA) complex cleaves both DNA strands via its two nuclease domains, generating a two-ended DNA double-strand break (DSB) in the protospacer sequence 5′ to the protospacer adjacent motif (PAM) sequence 5′-NGG-3′[1–3]. When donor DNA templates, including DNA sequences homologous to the target gene, are present, homology-directed repair (HDR), typically homologous recombination (HR), allows cells to incorporate fragments of the donor DNA template into the genome[1–3].

Although genome-editing technologies can correct gene mutations, DNA damage threatens genomic stability. DSB-specific error-prone repair pathways, such as non-homologous end joining (NHEJ) and microhomology-mediated end joining (MMEJ), can cause short nucleotide insertions or deletions (indels), long genomic deletions, large-scale chromosomal structural abnormalities, and chromothripsis[3,4]. Mutations can occur at both on- and off-target sites of CRISPR/Cas9.

[1]Department of Bioregulation and Cellular Response, Graduate School of Medicine, Osaka University, Suita, Osaka 565-0871, Japan. [2]Department of Genome Medicine, Tokyo Metropolitan Institute of Medical Science, Tokyo 156-0057, Japan. [3]Department of Genetics, Research Institute of Environmental Medicine (RIeM), Nagoya University, Nagoya 464-8601, Japan. [4]Department of Human Genetics and Molecular Biology, Nagoya University Graduate School of Medicine, Nagoya 464-8601, Japan. [5]Genomedia Inc., Tokyo 113-0033, Japan. [6]Institute for Advanced Co-Creation Studies, Osaka University, Suita, Osaka 565-0871, Japan. ✉e-mail: snakada@bcr.med.osaka-u.ac.jp

Furthermore, the random integration of exogenous DNA used as donor templates or for CRISPR/Cas9 expression can cause unintended genomic alterations, posing risks to therapeutic DSB- and exogenous DNA-dependent gene editing applications.

To address these risks, we developed a strategy for correcting heterozygous mutations by promoting interhomolog repair/recombination (IHR) using Cas9[D10A] nickase, which generates DNA nicks instead of DSBs. Nicks are DNA damage where a phosphodiester bond between adjacent nucleotides is absent in one strand of the DNA double helix, and they are quickly and accurately repaired by DNA single-strand break repair (SSBR) pathways[5]. Paired nick-induced DSBs (pnDSBs) created by nickase-induced nicks with <100 base pair (bp) offsets on both DNA strands efficiently create deletions at on-target sites but significantly reduce indel frequencies at off-target sites[6]. Nicks are occasionally converted into one-ended DSBs during replication. However, these one-ended DSBs are primarily repaired via HR or fork reversal rather than NHEJ or MMEJ[7–9]. Consequently, it can be anticipated that when nickase is used to introduce nicks on only one strand of the DNA double helix, the occurrence of unintended mutations at either on-target or off-target sites will be infrequent.

There are two types of IHR: interhomolog end-joining (IH-EJ) and interhomolog homologous recombination (IH-HR)[10]. IH-EJ joins DNA ends on homologous chromosomes, leading to an exchange of chromosome arms that can potentially alleviate the disease phenotype by converting two alleles with compound heterozygous mutations to a single wild-type (WT) allele and another allele with two mutations[10]. IH-HR repairs DNA breaks via HR using a homologous chromosome as a repair template and can correct single heterozygous mutations or one of the compound heterozygous mutations in the WT sequence.

Emerging evidence suggests that HDR can correct nicks[11–18], and several studies, including our own, have demonstrated that introducing multiple nicks (MNs) on both the target gene in a genome and donor plasmids enhances the HR between the genome and plasmids compared to a single nick (SN) on the target gene alone[19–23]. A recent report showed a frequent reciprocal exchange of chromosomal arms at DSBs on homologous chromosomes by IH-EJ in POLQ-depleted cells[10], suggesting that damaged homologous chromosomes may come into contact more often than expected. Furthermore, an allele-specific nick generated by nickase was found to induce efficient IH-HR in *Drosophila* somatic cells[24]. These findings prompted us to explore whether or not MNs on homologous chromosomes could stimulate IH-HR and thereby correct heterozygous mutations in human somatic cells.

In the present study, we discovered that MNs on both homologous chromosomes, including the mutant allele-specific nick adjacent to the mutation, can correct heterozygous mutations, such as small indels, missense mutations, and large deletions exceeding 600 bp, mainly via MN-induced IH-HR (MN-IH-HR) in human somatic cells. Consequently, we developed a method for correcting heterozygous mutations utilizing Cas9 nickase-induced MNs and a homologous chromosome as an endogenous repair template, named NICER.

## Results

### MNs on homologous chromosomes can induce IHR

TK6(IVGT) is a p53-positive near-diploid (47,XY,+13,der(14)t(14;20)(q32.3,q11.2),der(21)t(3;21)(q22;p11.2)) human lymphoblast cell line harboring a loss-of-function frameshift mutation caused by a 1-bp duplication (c.231dupC) in exon 4 of allele A and a neutral (gene function-preserving) 1-bp duplication (c.640dupG) in exon 7 of allele B in the thymidine kinase 1 (TK1) gene on the long arm of chromosome 17 (Fig. 1a)[25].

TK6261 cells were established by introducing a loss-of-function frameshift mutation (c.311_318del) in exon 5 of allele B using CRISPR/

Cas9 technology (Fig. 1a). Fluorescence in situ hybridization (FISH) showed that TK6261 cells contained two copies of the TK1 gene (Fig. 1b). TK1 is a pyrimidine salvage pathway enzyme that phosphorylates thymidine, converting it to thymidine monophosphate. Therefore, when aminopterin is used to block de novo DNA synthesis, TK6261 cells do not proliferate, even when hypoxanthine and thymidine are supplied. With TK1 restoration via genome editing, TK6261 cells can proliferate in 2′-deoxycytidine-hypoxanthine-aminopterin-thymidine medium (CHATM). Using this feature, we determined the TK1 activity recovery rate using colony formation or proliferation assays (Supplementary Figs. 1 and 2). When the mutation in exon 4 is targeted for correction, TK1 activity can be restored via gene correction of the exon 4 mutation, reciprocal exchange of chromosome arms between homologous chromosomes in intron 4, or $3n + 1$-bp ($n$: integer) deletion in exon 4 (reading frame recovery) of allele A (Fig. 1c).

First, we designed two sgRNAs, sgEx4_mt20s (in which the PAM sequence ended at the 20th nucleotide in the mutated exon 4, and the sequence was complementary to the sense strand), which specifically targeted the mutant sequence of the TK1 gene exon 4 on allele A with Cas9 (Supplementary Fig. 3a), and sgS0, which is complementary to the sense strand of intron 3 of the TK1 gene intron 3 on alleles A and B (Fig. 1d). The introduction of a single DSB by Cas9 and sgEx4_mt20s restored TK1 activity at a rate of 14.0% ± 2.17% in the colony formation assay (Fig. 1e, left panel). Among 82 single-cell-derived clones (SCCs) established in CHATM, only 3 (3.66%) exhibited WT exon 4; 78 (95.1%) SCCs showed a 1-bp deletion (c.234delC, c.235delT, c.236delG, or c.238delC), leaving the c.231dupC mutation in exon 4 (Fig. 1f and Supplementary Fig. 3b).

When TK6261 cells were electroporated with Cas9[D10A] nickase mRNA (Nickase) and either sgEx4_mt20s or sgS0, TK1 activity recovered at low rates. Multiple nicking (MNing) by sgEx4_mt20s (primary sgRNA; mutation-specific), gS0 (secondary sgRNA), and Nickase restored TK1 activity with approximately 17-fold higher efficiency than single nicking (SNing) by sgEx4_mt20s and Nickase (Fig. 1e, right panel). All 205 SCCs established in CHATM after MNing contained the WT exon 4 (Fig. 1f). Amplicon-based next-generation sequencing (AmpNGS) revealed that 98.9% ± 1.1% ($n = 3$) of all reads showed WT exon 4 in the cell populations with recovered TK1 activity (Supplementary Fig. 3c). Cell populations grown in CHATM after MN induction recovered the TK1 protein expression (Fig. 1g). Thus, MNs stimulate gene correction via MN-induced IHR (MN-IHR).

### More nicks improved the efficiency of gene correction via MN-IHR

We designed 67 additional sgRNAs complementary to the sense strand of the TK1 gene, spanning ±8.6 kb from the c.231dupC mutation (Supplementary Data 1). Each sgRNA (secondary sgRNA) was independently introduced into TK6261 cells using sgEx4_mt20s (primary sgRNA) and Nickase (Fig. 2a). TK1 activity recovery rates (calculated as proliferation in CHATM index [PCI]; see Methods section) were measured using a proliferation assay (Supplementary Figs. 1b and 2). Most MNs (offset = 24–8675 bp) restored TK1 activity >10-fold more efficiently than an SN created using Nickase, sgEx4_mt20s, and the non-targeting control sgRNA[26] (sgNTC) (Fig. 2a). Many MNs restored TK1 activity to the same level or better than that achieved with pnDSBs created using Nickase, sgEx4_mt20s, and sgEx4_88as (Fig. 2a). Furthermore, there was no correlation between the PCI and the offset range (Supplementary Fig. 3d). AmpNGS revealed that 96.0–99.5% of all reads contained the WT exon 4 sequences of the TK1 gene among 16 cell populations grown in CHATM after MNing (Fig. 2b).

Six sgRNAs with a high PCI in the proliferation assay (Fig. 2a) were employed as secondary sgRNAs in the colony formation assay (Fig. 2c, d). MNs generated using Nickase, sgEx4_mt20s, and 1 of the 6 sgRNAs produced TK1 activity-positive cells with an efficiency of 3–4% without

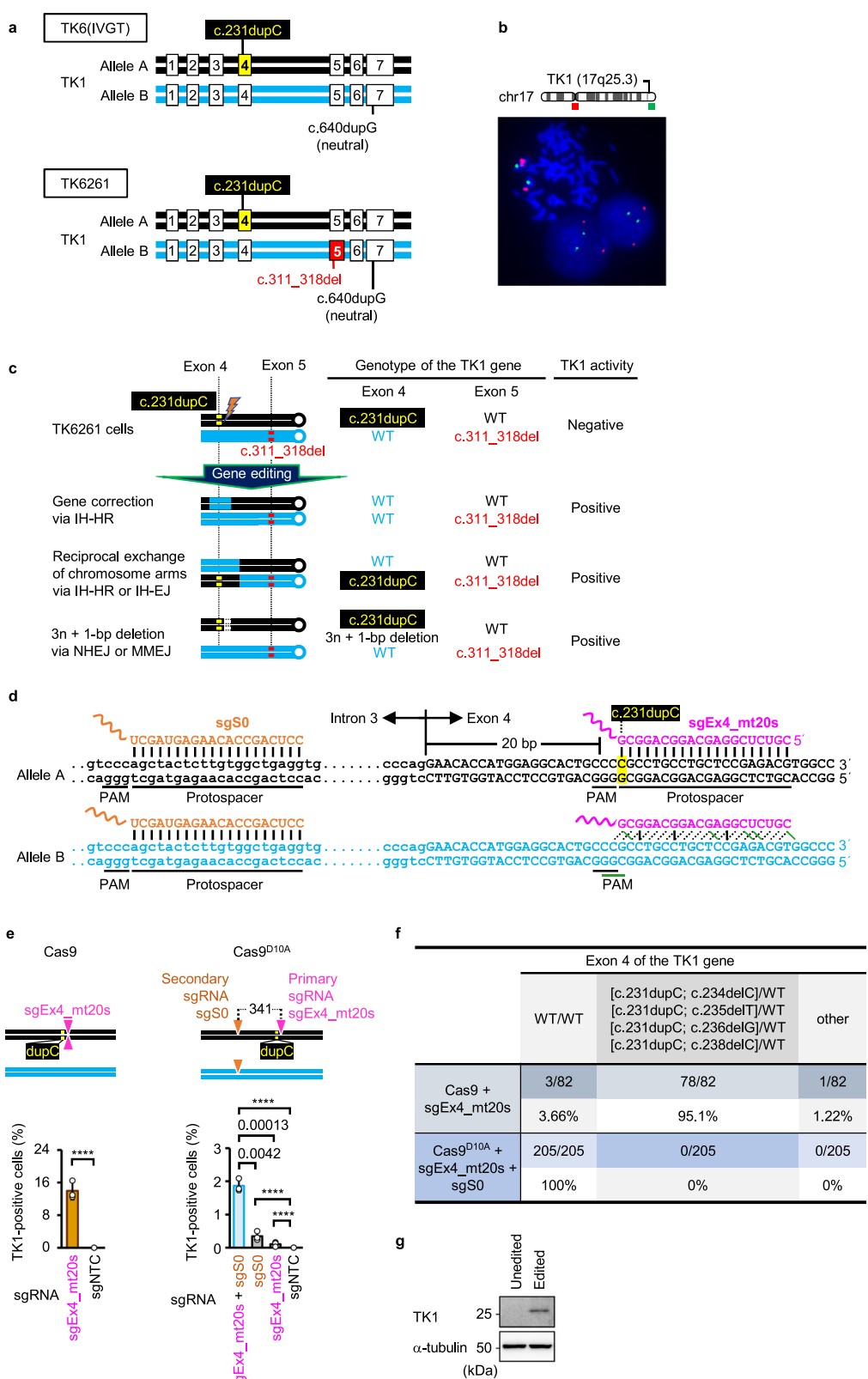

any selection (Fig. 2d, blue bars). All combinations achieved significantly higher gene correction efficiencies than sgEx4_mt20s and sgNTC. The TK1 activity was efficiently recovered, regardless of the offset size or whether the secondary nick was at 5′ or 3′ relative to the primary nick (Fig. 2d). Combinations of 2 of the 6 sgRNAs and sgEx4_mt20s with Nickase achieved 5–6% editing efficiency (Fig. 2d, pink bars).

## MNs on either the sense strand or the antisense strand corrected heterozygous mutations

We developed sgEx5_mt-4s, which specifically detects the mutated exon 5 sequence using Cas9 (Supplementary Fig. 4a–c). The MNs were introduced via sgEx5_mt-4s and either sgEx7_105s or sgS12. Combining these two sgRNAs improved the efficiency of TK1 activity recovery (Supplementary Fig. 4d). AmpNGS confirmed that cells grown in

**Fig. 1 | Strategy for analyzing IHR efficiency. a** A schematic illustration of heterozygous mutations in the TK1 gene in TK6(IVGT) and TK6261 cells. **b** Representative FISH image of the TK6261 cells. Red and green probes specifically hybridized to the centromeric regions of chromosomes 17 and 17q25.3, respectively. **c** A schematic illustration of the anticipated gene alterations that restore TK1 activity in TK6261 cells. **d** A schematic illustration of the genomic DNA sequence and part of the sgEx4_mt20s or sgS0 sequence. The (candidate) PAM and protospacer sequences are underlined. Possible hydrogen bonds between genomic DNA and sgRNAs are represented by vertical solid black lines, diagonal dashed black lines, or diagonal solid green lines. **e** Percentage of TK1 activity-positive cells in the colony formation assay (mean ± SD from three independent experiments). TK6261 cells were electroporated with either Cas9 mRNA (left panel) or Nickase (right panel), accompanied by the respective sgRNAs indicated in graphs. sgNTC refers to non-targeting control sgRNA.

In the schematic illustration (top panels), a combination of triangles pointing up and down denotes a DSB site created by Cas9, whereas inverted triangles indicate nick sites. dupC, c.231dupC. The proportion data were adjusted using the following formula: $AdX = (X + 10^{-10})/(1 + 2 \times 10^{-10})$, where X is the original proportion and AdX is the adjusted proportion. The adjusted proportions were transformed into a continuous scale using probit transformation and subjected to a two-tailed unpaired $t$-test (left panel, $t = 95.75$, $df = 4$) or one-way ANOVA followed by Tukey's multiple comparisons test (right panel, two-sided, $F$ [3, 8] = 475.1). Exact $P$-values are shown in the graphs. \*\*\*\*$P < 0.0001$. **f** Sanger sequencing for exon 4 of the TK1 gene in TK1 activity-positive SCCs. **g** TK1 protein expression in TK6261 cells. Unedited refers to TK6261 cells cultured in SCM without gene editing. Edited refers to TK6261 cells electroporated with sgEx4_mt20s, sgS0, and Nickase and then cultivated in CHATM. Data represent three independent experiments. Source data were provided as a Source Data file.

CHATM after MNing underwent correction of the exon 5 mutation (Supplementary Fig. 4e).

TSCER2 cells, which originate from TK6 cells, harbor multiple mutations, including a loss-of-function missense mutation (c.326G>A) in exon 5 of allele B and a c.231dupC mutation in allele A[27] (Fig. 3a). To introduce MNs onto the antisense strand of the TK1 gene using Nickase, we designed four sgRNAs that were complementary to the antisense strand of the TK1 gene (Fig. 3b). sgEx5_mt28as/TR2 with Cas9 specifically abrogated the mutated exon 5 sequence of the TK1 gene (Supplementary Fig. 4f, g). SNing using sgEx5_mt28as/TR2 rarely restored TK1 activity. However, introducing additional nicks in exon 7 of both alleles increased the efficiency by 44.1-fold (Fig. 3c).

### SNs on both homologous chromosomes induced IHR

In TK6261 cells, we utilized a base editor (BE) and sgS43 to introduce a four-base substitution on either allele A (TK6261_S43v44 cells) or allele B (TK6261_S43v11 cells) (Fig. 3d and Supplementary Fig. 5a). MNing with sgEx4_mt20s and substituted sequence-specific sgS43_mt0 (Supplementary Fig. 5b, c) restored TK1 activity exclusively in TK6261_S43v11 cells (Fig. 3e). These data indicate that MN-IHR is effective when nicks are present in both alleles. Similar results were obtained in different experimental settings. Introducing an SN on each allele by co-introduction of sgEx4_mt20s and sgEx5_mt-4s, but not sgEx5_WT-4s, with Nickase efficiently restored TK1 activity (Fig. 3f, g, Supplementary Fig. 6). This efficiency was greater than the sum of the individual effects of sgEx4_mt20s and sgEx5_mt-4s, indicating a synergistic effect of MNing.

### MNs promoted gene conversion via IH-HR

After introducing MNs, we established SCCs in standard culture medium (SCM) and performed Sanger sequencing of exons 4 and 5 of the TK1 gene as well as long-range polymerase chain reaction (PCR) to amplify regions, including the target sites of sgRNAs for 94 SCCs in 3 independent experiments. The TK1 activity was assessed by culturing SCCs in CHATM (Supplementary Fig. 7a–c). All of these data are summarized in Supplementary Fig. 8a.

A few SCCs exhibited long deletions. All TK1 activity-positive SCCs contained the WT/WT exon 4 and WT/c.311_318del exon 5 (Supplementary Fig. 8a). We did not detect any cells with c.231dupC/c.231dupC exon 4. Furthermore, no TK1 activity-positive clones showed WT/c.231dupC exon 4 and WT/c.311_318del exon 5 concomitantly (Supplementary Fig. 8a). These findings strongly suggest that the recovery of TK1 activity and correction of exon 4 mutations are mediated mainly by IH-HR rather than by IH-EJ.

Among SCCs established after introducing two nicks on both homologous chromosomes by sgEx4_mt21/WT20s (Supplementary Fig. 8b, c), sgS14, and Nickase, we detected some SCCs with WT/WT exon 4 and others with c.231dupC/c.231dupC exon 4 (Supplementary Fig. 8a). Of the 282 SCCs established after introducing an SN on each allele using sgEx4_mt20s, sgEx5_mt-4s, and Nickase, 4 showed TK1 activity and contained either WT/WT exon 4 with heterozygous exon 5

or WT/WT exon 5 with heterozygous exon 4. The three SCCs contained WT/WT exons 4 and 5 (Supplementary Fig. 8d). No SCC contained c.231dupC/c.231dupC exon 4 or c.311_318del/c.311_318del exon 5. These data suggested that the allele carrying the nick serves as the recipient allele for MN-IH-HR in the region surrounding the nick.

### A nick on the donor allele both extended and restricted the region incorporated from the donor allele to the recipient allele

To investigate the correlation between the length of the region incorporated from the donor allele to the recipient allele and the physical positions of the MNs, we used sgEx5_mt-4s to introduce the primary nick adjacent to the mutation in exon 5 and the secondary nicks at regions located 3′ (centromeric) relative to the primary nick (Fig. 4a). We performed AmpNGS in TK6261 cell populations grown in CHATM after introducing MNs or an SN, and determined the percentage of reads with the WT sequence of exons 5 or 7. In cells with restored TK1 activity after SNing, the proportion of WT exon 7 reads was approximately 50%. Introducing secondary nicks at sites located 3′ to the neutral duplication in exon 7 led to a significant increase in the percentage of WT exon 7 reads (Fig. 4b). In contrast, introducing secondary nicks between the mutation in exon 5 and the neutral mutation in exon 7 resulted in approximately 50% of the WT exon 7 reads (Fig. 4b).

We generated TK6261_int5v32 cells harboring a heterozygous G>A single nucleotide variant (SNV) in intron 5 of the TK1 gene on allele B by electroporating BE and sgEx6_-92s in TK6261 cells (Fig. 4c, Supplementary Fig. 9a). After electroporation with sgEx5_mt-4s, sgS12, and Nickase, we established 714 SCCs in the SCM. Among the 11 SCCs containing WT exon 5, 2 had deletions between the nicks (Supplementary Fig. 9b). Of the remaining nine clones, seven exhibited heterozygous exon 4, WT intron 5, and WT exon 7 (Fig. 4d), suggesting that the DNA sequence between the nicks on allele B was frequently replaced with that of allele A. Long-range PCR and topoisomerase-based (TOPO) cloning confirmed these results (Supplementary Fig. 9c).

We generated TK6261_S62v49 cells harboring a heterozygous SNV (G>A) in intron 3 of the TK1 gene on allele A using a BE and sgS62 (Fig. 4e). The exon 4 mutation was corrected using sgEx4_mt20s as the primary sgRNA. Introducing secondary nicks 5′ (telomeric) to the SNV resulted in 80.6% ± 2.5% of reads showing the WT sequence in intron 3 (Fig. 4f). Secondary nicks 3′ (centromeric) to the SNV did not increase the percentage of WT reads efficiently, but 65.9% ± 2.1% or 67.0% ± 3.8% (Fig. 4f). Thus, secondary nicks are crucial for extending and restricting the region incorporated from the donor allele to the recipient allele in the 3′ or centromeric direction and more flexibly in the 5′ or telomeric direction.

We investigated whether or not NICER could repair deletions spanning a few hundred base pairs in TK6261_LD407E cells, which harbor a 662-bp deletion and a 1-bp insertion in allele A. sgLD407E_-4s and sgLD407E_6s were designed to target the breakpoint of allele A (Fig. 4g and Supplementary Fig. 10a, b). Combining Nickase and sgS9 with one of these sgRNAs efficiently rescued the TK1 activity, whereas

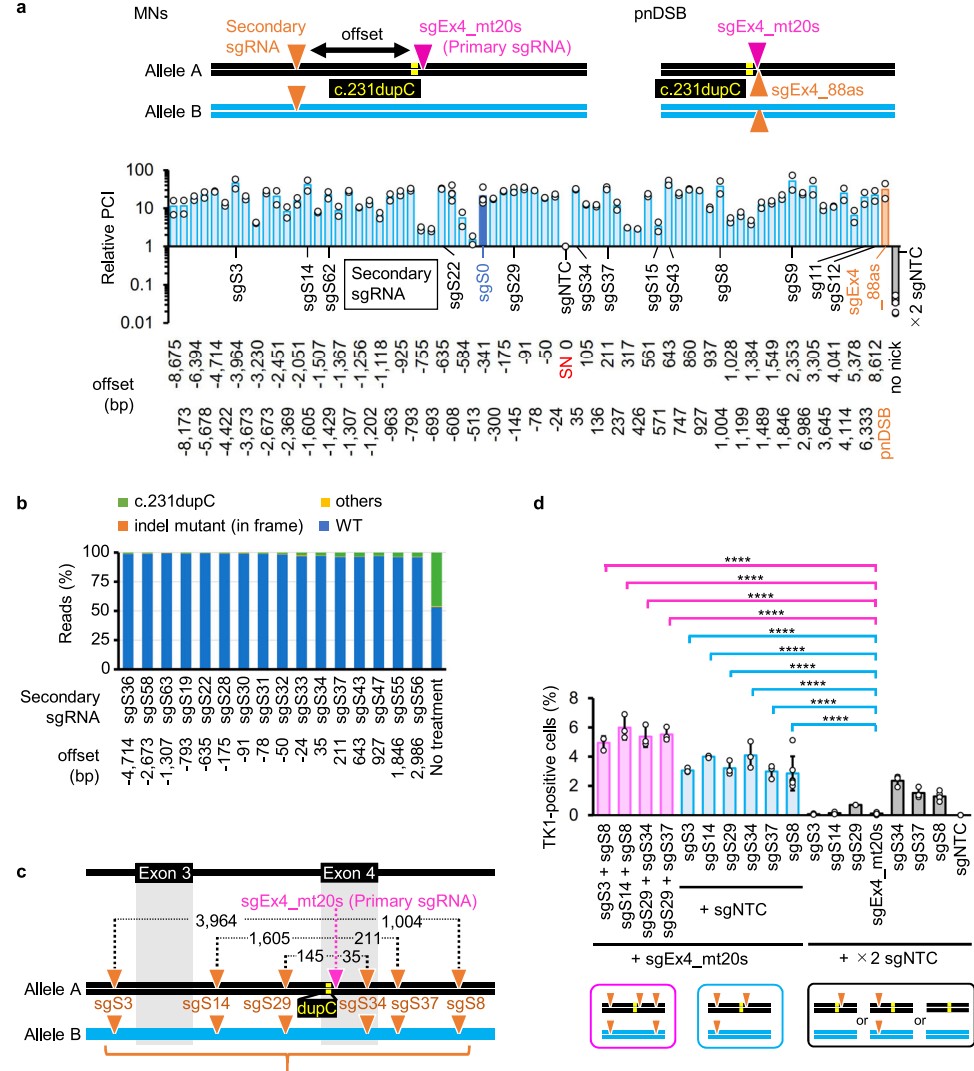

**Fig. 2 | MNs efficiently induce IHR. a** Gene correction efficiency was determined using a proliferation assay. Relative PCIs (1 = average PCI when TK6261-EGFP cells were electroporated with sgEx4_mt20s, sgNTC, and Nickase [an SN was generated in this procedure], indicated as offset = 0) were calculated (mean of four independent experiments for sgS0 [offset = −341], sgS6 [offset = −608], sgNTC [SN only, offset = 0], and ×2 sgNTC [no nick]; mean of two independent experiments for the others). The vertical axis represents a logarithmic scale. The name and sequence of each sgRNA, along with the offset, are provided in Supplementary Data 1. Light blue: MNs were generated on the sense strand; dark blue: MNs were generated via sgEx4_mt20s + sgS0 (used as the standard); orange: sgEx4_mt20s + sgEx4_88as (pnDSB); gray: no nick; sgNTC: non-targeting control sgRNA; MNs: multiple nicks; pnDSB: paired nick-induced DSB. **b** AmpNGS of exon 4 of the TK1 gene. Percentage of reads indicating WT, c.231dupC, $3n + 1$-bp ($n$: integer) deletion with c.231dupC, or other types of exon 4 sequences are indicated. TK6261 cells were electroporated

with Nickase and sgRNAs. TK1 activity-positive cells were enriched in CHATM and subsequently subjected to AmpNGS. No treatment: cells were cultivated in SCM and not gene-edited. **c** A schematic illustration of mutations and nicked sites (indicated by inverted triangles). dupC, c.231dupC. **d** Percentage of TK1 activity-positive cells determined in a colony formation assay (mean ± SD from nine [sgEx20s_mt20s + ×2 sgNTC and sgNTC + ×2 sgNTC], six [sgEx20s_mt20s + sgS8 + sgNTC and sgS8 + ×2 sgNTC], and three [others] independent experiments). The proportion data were adjusted using the following formula: AdX = (X + 10^{−10})/(1 + 2 × 10^{−10}), where X is the original proportion and AdX is the adjusted proportion. The adjusted proportions were transformed into a continuous scale using probit transformation and subjected to a one-way ANOVA followed by Dunnett's multiple comparisons test (two-sided). $F_{(17, 54)} = 49.46$. ****$P < 0.0001$. Source data were provided as a Source Data file.

combining Cas9 mRNA with one of these sgRNAs was unsuccessful (Fig. 4h). Long-range PCR of 94 × 3 SCCs established in CHATM after MNing revealed that over 83.3% ± 3.3% did not contain a long deletion (Fig. 4i). Among 228 long-deletion-negative PCR products, 226 had the WT sequence at both breakpoints in exon 4 and intron 4, confirming that MNing effectively and precisely corrected the long deletions.

To investigate whether or not MNing induces loss of heterozygosity (LOH) distant from the target gene, we analyzed the proportion of SNV (G>A) at chr17:78279901 (GRCh38/hg38) situated 17 kb telomeric to the c.231dupC mutation. AmpNGS revealed that TK1 activity-positive TK6261 cell populations had slightly increased

percentages of "G" at chr17:78279901 (Supplementary Fig. 11a–c). Sanger sequencing showed a few SCCs established in SCM after introducing MNs or a pnDSB exhibited "G" or "A" homozygosity or hemizygosity at chr17:78279901. Interestingly, all 282 (94 × 3) SCCs established after introducing an SN on each allele electroporating sgEx4_mt20s, sgEx5_mt-4s, and Nickase exclusively contained "G/A" at chr17:78279901 (Supplementary Fig. 11d).

## Mechanistic insights into MN-IH-HR

HR for DSB repair is typically limited to the late S and G2 phases of the cell cycle. CtIP and MRE11 initiate HR by resecting the DSB ends. After

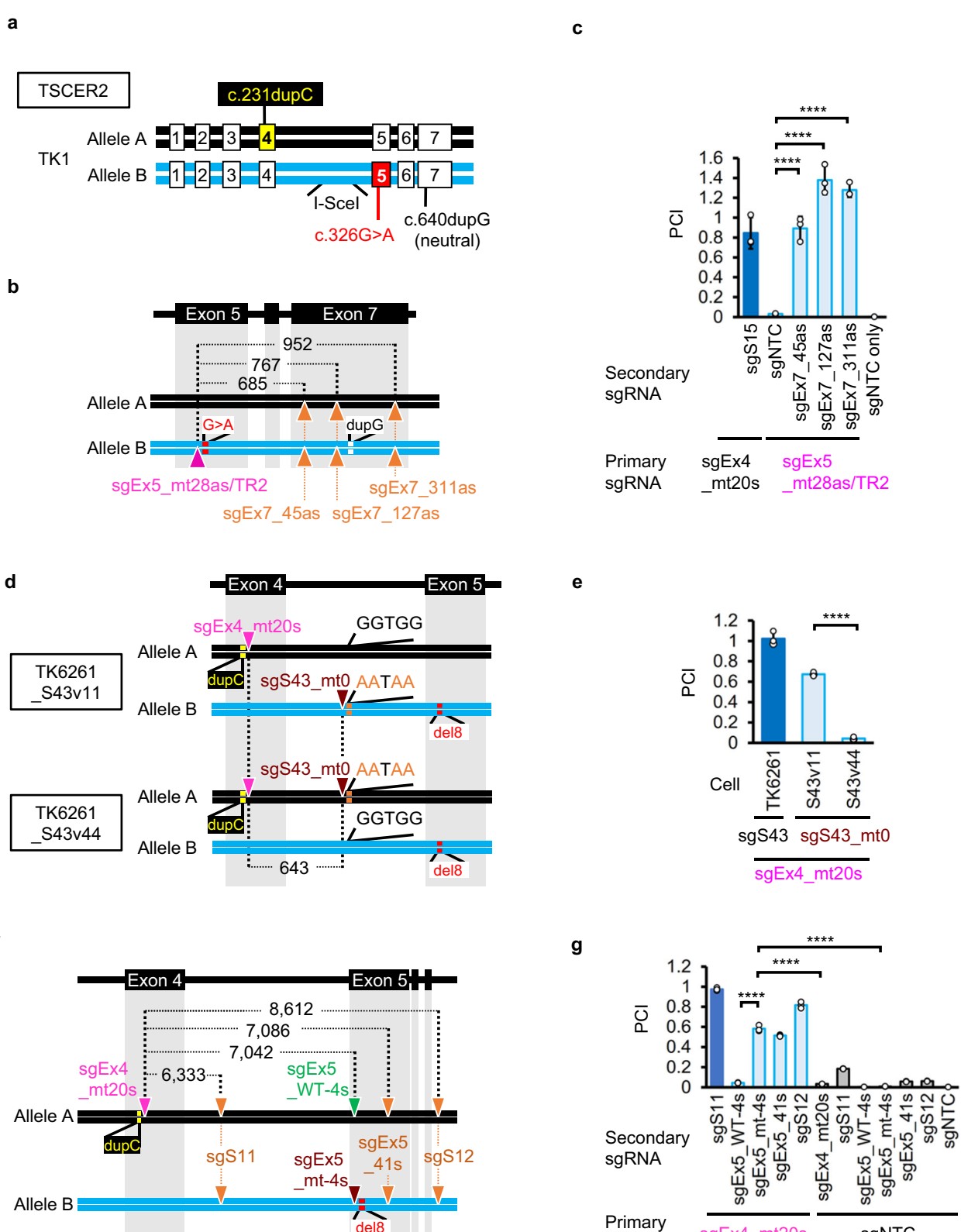

cutting the 5'-terminated strand by MRE11 endonuclease activity, MRE11 (3' to 5' direction) and EXO1/BLM/DNA2 (5' to 3' direction) resect the strand to create a 3' single-stranded DNA (ssDNA) overhang[28]. The exposed ssDNA is protected using RPA coating. BRCA1 plays multiple roles in DSB HR, including promoting DNA end resection and recruiting PALB2-BRCA2 to the DSB sites[29]. BRCA2 exchanges

RPA with RAD51 on ssDNA, after which strand invasion of the RAD51-ssDNA filament into the homologous DNA helix occurs[29].

Palbociclib, a specific CDK4 and CDK6 inhibitor[30], arrested TK6261 cells in the G1 phase after 24 h of treatment. The removal of palbociclib released cells from G1 arrest (Supplementary Fig. 12a, b). The recovery efficiency of TK1 activity diminished when TK6261 cells

**Fig. 3 | Efficient MN-IHR occurs with nicks on both homologous chromosomes.** Gene correction efficiency was evaluated using a proliferation assay. **a** A schematic illustration of compound heterozygous mutations in the TK1 gene in TSCER2 cells. **b**, **c** A missense mutation in exon 5 was corrected by introducing MNs on the antisense strand of the TK1 gene in TSCER2 cells. **b** A schematic illustration of mutations and nicked sites (indicated by triangles). **c** Mean PCI ± SD from three independent experiments. The standard sample was TK6261_EGFP cells electroporated with sgEx4_mt20s, sgS15, and Nickase (dark-blue bar). The test samples were TSCER2 cells electroporated with the indicated sgRNAs and Nickase. **d**, **e** The mutation in exon 4 was corrected by either SNs on both alleles or two nicks on allele A. **d** A schematic illustration of mutations and nicked sites (inverted triangles). **e** Mean PCI ± SD from three independent experiments. The standard sample was TK6261_mCherry cells electroporated with sgEx4_mt20s, sgS43, and Nickase (dark-

blue bar). The test samples were TK6261, TK6261_S43v11, and TK6261_S43v44 cells electroporated with the indicated sgRNAs and Nickase. **f**, **g** Mutations in exon 4 and/or 5 of the TK1 gene in TK6261 cells were corrected by introducing MNs. **f** A schematic illustration of mutations and nicked sites (inverted triangles). **g** Mean PCI ± SD from three independent experiments. The standard sample was TK6261_mCherry cells electroporated with sgEx4_mt20s, sgS11, and Nickase (dark-blue bar). The test samples were TK6261_EGFP cells electroporated with the indicated sgRNAs and Nickase. dupC, c.231dupC; del8, c.311_318del; G>A, c.326G>A; dupG, c.640dupG. I-sceI is the I-sceI recognition sequence. Data were analyzed using a one-way ANOVA followed by Dunnett's multiple comparison test (two-sided). $F_{(5, 12)} = 98.19$ (**c**), $F_{(2, 6)} = 446.1$ (**d**), or $F_{(11, 24)} = 1909$ (**g**). ****$P < 0.0001$. Source data were provided as a Source Data file.

were treated with palbociclib from one day prior to three days post-electroporation with sgRNAs and Nickase. However, it maintained a 56% or 35% reduction compared to when TK6261 cells were not treated with palbociclib or were treated with palbociclib only prior to electroporation, respectively (Fig. 5a, b). Sanger sequencing of TK1 activity-positive cell populations established after MNing the TK1 gene in palbociclib demonstrated that the exon 4 mutation of the TK1 gene had been completely corrected (Supplementary Fig. 12c). These findings suggest that MN-IH-HR occurs even in the G1 phase.

Chromatin immunoprecipitation (ChIP) assays were conducted using anti-RPA or RAD51 antibodies, followed by droplet digital PCR. Neither RPA nor RAD51 accumulation in the region surrounding the Cas9-sgEx4_mt20s complex-generated DSB was detected (Fig. 5c–e), suggesting that most Cas9-induced DSBs are repaired by NHEJ. RPA accumulation was scarcely detected when an SN was generated on one allele using the sgEx4_mt20s-Cas9[D10A] complex (Fig. 5d). When the sgS14-Cas9[D10A] complex was employed to introduce SNs on both alleles, RPA accumulation was observed in regions III and II (between the target sites of sgS14 and sgEx4_mt20s, Fig. 5c) but not in region I (near the target site of sgEx4_mt20s, Fig. 5c) (Fig. 5d). When MNs were generated by sgEx4_mt20s, sgS14, and Nickase, RPA accumulation was detected not only at the nicked sites (regions I and III) but also in the regions between the two target sites (region II) (Fig. 5d). Similar results were obtained for RAD51 accumulation (Fig. 5e). These findings suggest that the generation of MNs leads to the exposure of single-stranded DNA in an extensive area surrounding the target sites and that MNs can be repaired via a RAD51-dependent pathway.

We assessed the impact of knockout or auxin-induced silencing of DNA repair-related genes on MN-IR-HR efficiency. We created MNs or a pnDSB by electroporating TSCER2 or TSCER2-derived cells with Nickase and sgRNA pairs sgEx4_mt20s and sgS14 or sgEx4_mt20s and sgEx4_88as, respectively. Cell populations grown in CHATM after pnDSB introduction in exon 4 exhibited 89.6% ± 1.2% of the WT exon 4 reads (Supplementary Fig. 3c), indicating that pnDSB restores TK1 activity via IHR and can serve as a positive control.

Auxin-induced silencing of auxin-induced degron (AID)-tag[31]-fused endogenous BRCA1 (BRCA1-AID[32]) or BRCA2 (BRCA2-AID) reduced the pnDSB-induced gene correction efficiency to <20%. However, knockdown of BRCA1-AID or BRCA2-AID only decreased the MN-IH-HR efficiency to approximately 50% of that observed in cells expressing BRCA1-AID or BRCA2-AID, respectively (Fig. 6a, b, Supplementary Fig. 13a, b). Knockout of EXO1[33] suppressed both pnDSB-mediated IHR and MN-IH-HR (Fig. 6c, Supplementary Fig. 13c). In contrast, auxin-induced knockdown of CtIP-AID[34] suppressed pnDSB-mediated gene correction but did not affect MN-IH-HR efficiency (Fig. 6a, Supplementary Fig. 13d). We used Cas9[H840A] to nick the antisense strand; however, the effect of CtIP knockdown was negligible or nonexistent (Supplementary Fig. 13e, f). BLM knockout[35] did not affect the MN-IH-HR efficiency (Supplementary Fig. 13g).

RAD54 enhances HR by promoting RAD51-mediated strand exchange and facilitating the migration and resolution of Holliday junctions (HJs)[36,37]. RAD54 knockout[34] reduced pnDSB-mediated gene correction but had less of an effect on MN-IH-HR efficiency (Fig. 6d). Knockout of MUS81[38], a key factor in HJ resolution[39], did not suppress MN-IH-HR (Supplementary Fig. 14a), similar to BLM, which plays a role in HJ dissolution.

Knockout of SMARCAL1[40], an annealing-dependent helicase that promotes branch migration and fork reversal[41], elevated MN-IH-HR efficiency (Fig. 6e). The mismatch repair system (MMR) affects genome editing efficiency using nickase-based techniques, such as base editing and prime editing. Thus, we investigated the involvement of MMR in MN-IH-HR. Knockout of MSH2[38], a mismatch sensor, increased the MN-IH-HR efficiency. In contrast, MLH1 knockout[38] had no impact, and MLH3 knockout[38] had a small impact on MN-IH-HR efficiency when the secondary sgRNA (sgS14) and nickase created a nick in a region 5′ to the target site of the primary sgRNA (Fig. 6e). Knockout of these MMR factors did not affect the pnDSB-mediated gene correction (Supplementary Fig. 14b). Knockout of XRCC1[42], which is essential for canonical nick repair[5], slightly increased MN-IH-HR frequency. Knockout of XPA[42], a nucleotide excision repair factor[43], did not affect MN-IH-HR efficiency (Supplementary Fig. 14a). Although POLQ depletion increased IH-EJ when both homologous chromosomes were damaged by DSBs or nicks[10], POLQ knockout did not affect the MN-mediated gene correction efficiency (Supplementary Fig. 14c, d).

Although Cas9-induced single-strand template repair (SSTR) requires the Fanconi anemia pathway, with genome editing efficiency via SSTR severely reduced in FANCA-depleted cells[44], a disruption of the FANCA gene in TK6261 cells (Supplementary Figure 14e, f) only partially reduced the MN-mediated gene correction efficiency (to approximately 75%) (Fig. 6f).

## MNs rarely generated short indels at the on- and off-target sites of CRISPR/Cas9

We analyzed the frequency of MN-induced short indels at on- or off-target sites using AmpNGS. As controls, we electroporated TK6261 cells with Cas9 mRNA and sgEx4_mt20s in either the presence or absence of a 500-nt-long single-stranded oligodeoxynucleotide (ssODN) to correct the exon 4 mutation.

Cas9-induced DSBs with or without ssODNs generated high levels of indels in SCM-cultured cells (Fig. 7a). Even after selecting TK1 activity-positive cells via CHATM culture, DSB-based gene editing exhibited high indel frequency (Fig. 7a). Conversely, the indel frequency generated by MNing was quite low in the regions surrounding the target sequence of the primary sgRNA (sgEx4_mt20s) and secondary sgRNA (sgS14, sgS34, or sgS8) (Fig. 7a–d).

Indel frequencies at sgRNA off-target sites predicted by in silico analyses (Supplementary Data 2–5) were also analyzed. At the most likely off-target site of sgEx4_mt20s, chr10:86963531–86963550 (GRCh38/hg38), Cas9 generated indels at a frequency of >84%, whereas all combinations of MNs generated indels at a frequency

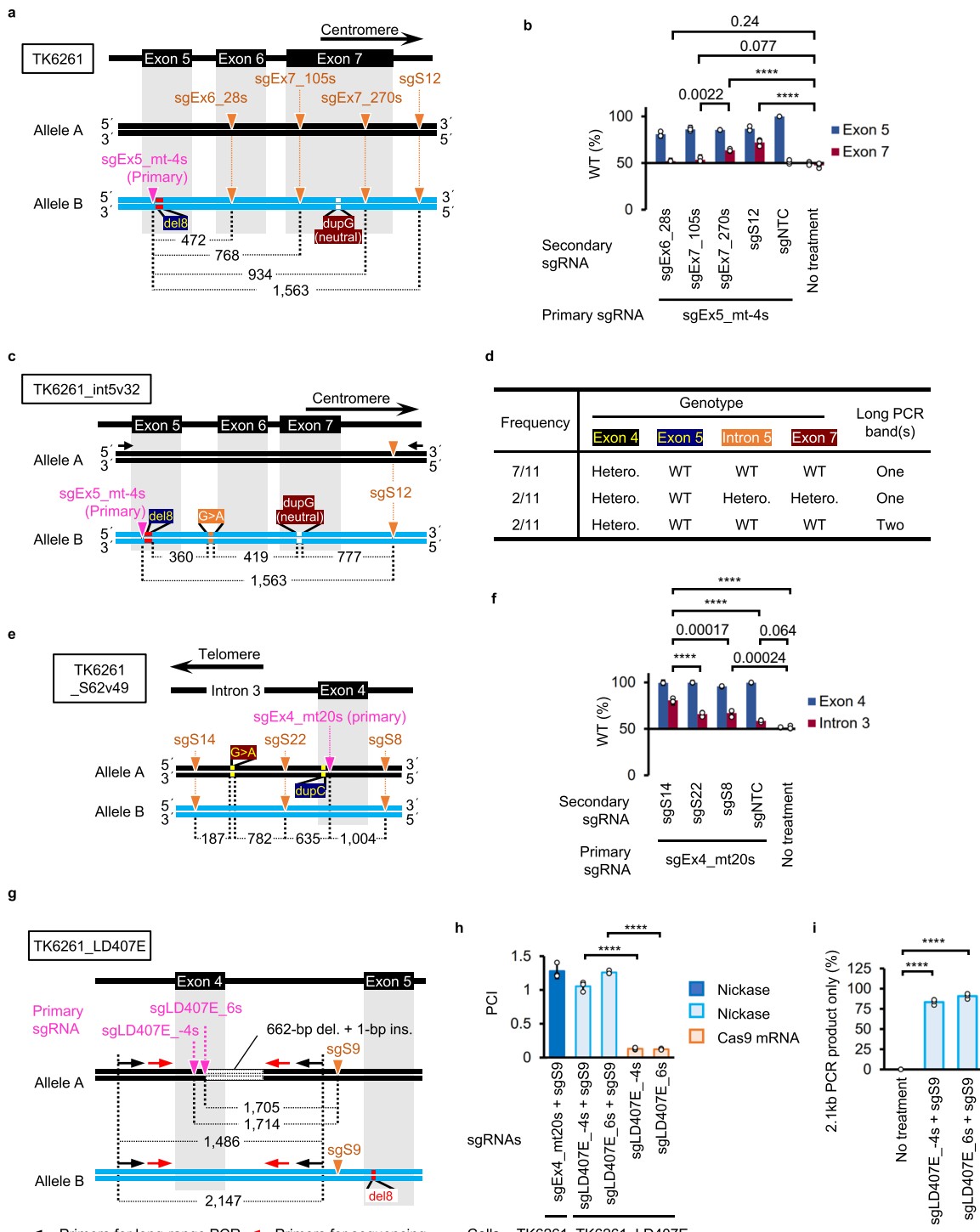

**Fig. 4 | Secondary nicks both extend and restrict the incorporation of DNA sequences from the donor allele to the recipient allele in MN-IH-HR.**
**a**, **c**, **e**, **g** Schematic illustrations of mutations, nicked sites, and primers. **a**, **b**, **e**, **f** Cell populations with TK1 activity were enriched in CHATM and subjected to AmpNGS for the regions surrounding exons 4 (**f**), 5 (**b**), and 7 (**b**) and intron 3 (the region surrounding the target site of sgS62, **f**) of the TK1 gene. **b**, **f** Percentage of reads with WT sequences (mean ± SD from three independent experiments). TK6261 (**b**) and TK6261_S62v49 (**f**) cells were electroporated with the indicated sgRNAs and Nickase. **c**, **d** Exon 4, intron 5, and exon 7 genomic DNA sequences and long-range PCR of 11 exon 5-corrected TK6261_int5v32-derived SCCs. **d** Frequency of SCCs with the indicated genotypes. Hetero.: heterozygote. **g**–**i** The long deletion in TK6261_LD407E cells was corrected by NICER. Mean PCI ± SD from three independent experiments. The standard sample was TK6261_mCherry cells electroporated with sgEx4_mt20s, sgS9, and Nickase (dark-blue). The test samples were

TK6261_LD407E cells electroporated with either Nickase or Cas9 mRNA, accompanied by the respective sgRNAs indicated in the graph (**h**). Percentage of TK1 activity-positive SCCs showing only 2.1-kb PCR products in long-range PCR (mean ± SD from three [94 clones/experiment] independent experiments). The proportion data were adjusted using the following formula: $AdX = (X + 10^{-10})/(1 + 2 \times 10^{-10})$, where X is the original proportion and AdX is the adjusted proportion (**i**). The original proportion data (**b**, **e**) and adjusted proportion data (**i**) were transformed into a continuous scale using probit transformation and subjected to a one-way ANOVA followed by Tukey's (**b**, **f**, **h**) or Dunnett's (**i**) multiple comparison test (two-sided). $F (5, 12) = 48.72$ (**b**), $F (4, 10) = 59.09$ (**f**), $F (4, 10) = 289.1$ (**h**) or $F (2, 6) = 3,129$ (**i**). Exact *P*-values are shown in the graphs (**b**, **f**). ****$P < 0.0001$. dupC, c.231dupC; del8, c.311_318del; dupG, c.640dupG; G>A, SNV at the target site of sgS62. Source data were provided as a Source Data file.

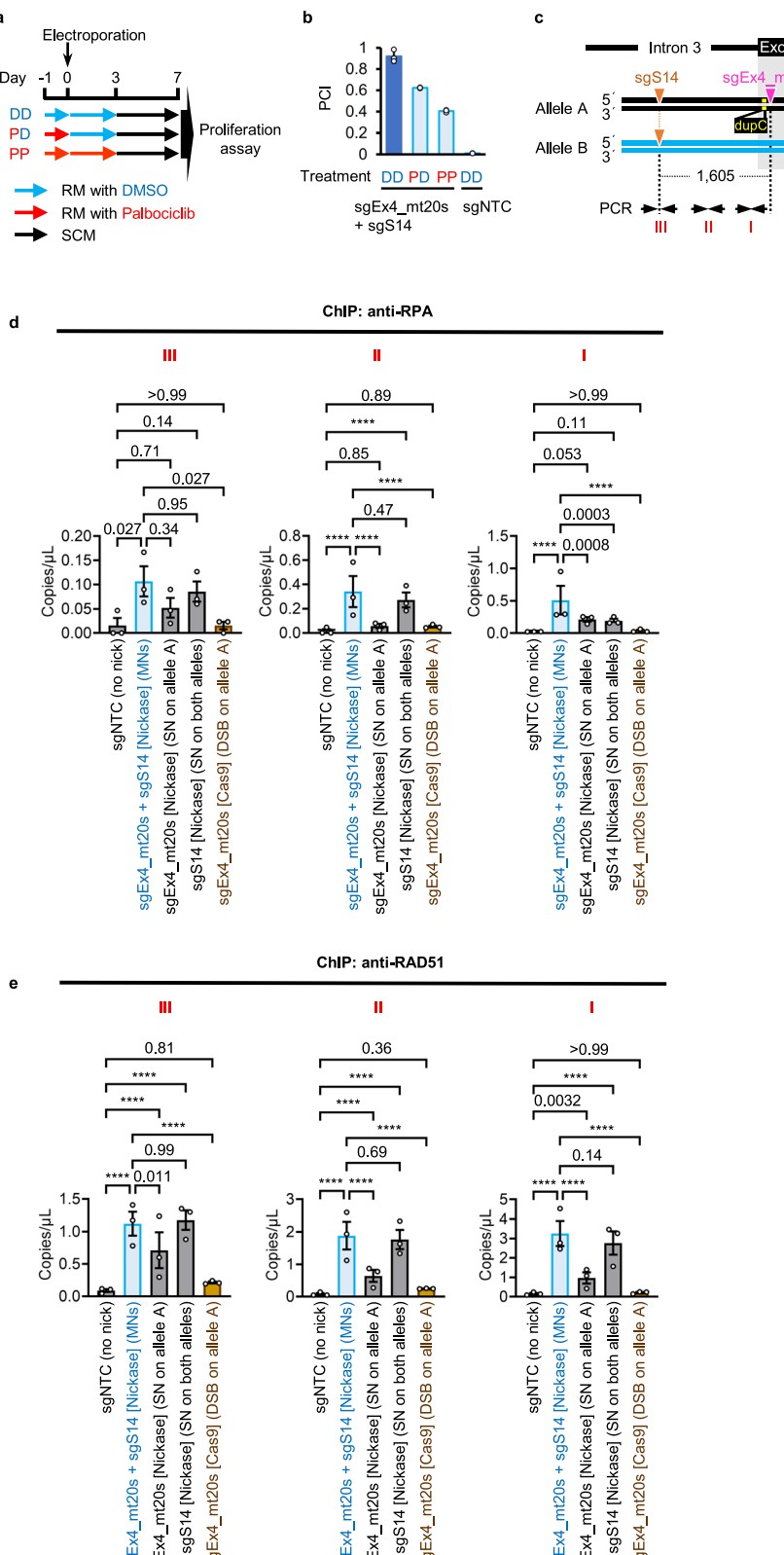

of ≤0.3% (Fig. 7e). Cas9 with sgS8 generated indels at a frequency of 92.0% ± 2.4% at the top candidate off-target site of sgS8, chr20:9270340–9270359 (GRCh38/hg38), whereas Nickase with sgEx4_mt20s and sgS8 did not generate indels at this site, even after enriching TK1 activity-positive cells in CHATM (Fig. 7f). Furthermore, MNs did not generate short indels at any of the off-target candidate sites analyzed (Supplementary Fig. 15).

## Genomic alterations in gene-corrected SCCs detected through whole-genome sequencing (WGS)

We performed WGS on TK6261 SCCs with corrected exon 4 of the TK1 gene and SCCs derived from untreated TK6261 cells. We compared the WGS results of each SCC and the reference and determined the number of genomic alterations using the Genome Analysis Toolkit (GATK) or Illumina's MANTA program. Because cell lines spontaneously generate

**Fig. 5 | MN-IH-HR efficiency in G1 arrested cells and accumulation of RPA and RAD51 induced by MNs. a, b** MN-IH-HR efficiency in G1-arrested cells. The mutation in exon 4 of the TK1 gene was corrected by NICER. **a** A flowchart illustrating the drug treatment procedure. RM, recovery medium. **b** Mean PCI ± SD from three independent experiments. The standard sample was TK6261_mCherry cells electroporated with sgEx4_mt20s, sgS14, and Nickase and cultured in RM with DMSO (dark-blue bar in **b**). The test samples were TK6261 cells electroporated with Nickase and either sgNTC or a combination of sgEx4_mt20s and sgS14, and cultured in RM with 4 μM palbociclib or DMSO. **c−e** A ChIP-droplet digital PCR assay. **c** Schematic illustrations of mutation and nicked sites (inverted triangles), along with PCR primers for droplet digital PCR (arrows). dupC, c.231dupC. **d, e** Results of droplet digital PCR following ChIP are shown. TK6261 cells were electroporated with either Nickase or Cas9 mRNA, accompanied by the respective sgRNAs indicated in graphs. One day post-electroporation, the cells were subjected to a ChIP assay using anti-RPA (**d**) or anti-RAD51 (**e**) antibodies. Droplet digital PCR was conducted using specific primers for regions I, II, or III, as indicated in the schematic illustration (**c**). Data represent the mean ± standard error of the mean (SEM) of triplicate measurements from three independent experiments. Data were analyzed using a two-way ANOVA followed by Tukey's multiple comparison test (two-sided). $F_{(4, 30)} = 17.7$ (I, **d**), 25.31 (II, **d**), 4.049 (III, **d**), 102.9 (I, **e**), 175.8 (II, **e**) or 37.29 (III, **e**). Exact $P$-values are shown in the graphs. ****$P < 0.0001$. Source data were provided as a Source Data file.

many genomic alterations during cell culture, we focused on the top 99 in silico-predicted off-target sites for each sgRNA (Supplementary Data 2–5). GATK analyses revealed that the combination of Cas9 mRNA, sgEx4_mt20s, and ssODNs generated SNVs or small indels at a frequency of $2.53 \times 10^{-2}$ to $3.00 \times 10^{-2}$ (from within ±10 bp to ±50 bp from the 3′ end of the off-target sequences)/off-target/SCC, while the combination of Nickase, sgEx4_mt20s, and sgS14, sgS8, or sgS34 did not generate SNVs or small indels within ±50-bp from the 3′ end of the off-target sequences during gene correction (Table 1). MANTA program analyses showed that gene correction using Cas9 mRNA, sgEx4_mt20s, and ssODN generated structural variants (SVs) at a frequency of $2.00 \times 10^{-3}$ to $4.00 \times 10^{-3}$ (from within ±10 bp to ±100 bp from the 3′ end of the off-target sequences)/off-target/SCC, whereas the combination of Nickase, sgEx4_mt20s, and sgS14 or sgS8 did not generate any SVs during gene correction (Table 2). The combination of Nickase, sgEx4_mt20s, and sgS34 generated only one SV in one of the nine analyzed SCCs.

## The gene correction rate was enhanced through repeated application of NICER

Given that DNA sequences remained unaltered in cells in which the mutation was not corrected by MN introduction (Fig. 7a–d), we hypothesized that repeated application of NICER could enhance gene correction efficiency. Electroporating cells with sgEx4_mt20s, sgS3, sgS8, and Nickase 3 times every 2 weeks restored TK1 activity in 18.5% ± 0.85% of cells, representing a 2.6-fold increase in efficiency compared to 1-time MN induction (Fig. 8a). Electroporation of cells with sgEx4_mt20s, sgS14, and Nickase 3 times every 4 days led to the restoration of TK1 activity in 13.6% ± 3.8% of cells, demonstrating a 2.3-fold higher efficiency than that achieved with 1-time MN generation (Fig. 8b).

## NICER-mediated gene correction in compound heterozygous diseases

AP39P cells, skin fibroblasts from a patient with aplastic anemia, mental retardation, and dwarfism syndrome, harbored compound heterozygous mutations in the alcohol dehydrogenase 5 (*ADH5*) gene (c.564+1G>A, c.832G>C; Fig. 8c) and a heterozygous mutation in aldehyde dehydrogenase 2[45,46]. We designed a sgRNA specifically targeting c.832G>C in exon 7 (sgAP39P_Ex7_mt7s; Supplementary Fig. 16a) and two sgRNAs targeting intron 6 or exon 7 of the ADH5 gene (Fig. 8c). NICER successfully restored ADH5 protein expression in AP39 cells (Fig. 8d).

Correcting gene mutations in patients with Fanconi anemia using CRISPR/Cas9 and ssODN is challenging due to the reliance on the Fanconi anemia pathway[44]. However, MN-IH-HR efficiency exceeded 75% in FANCA-mutated cells compared to WT FANCA-expressing cells (Fig. 6f), suggesting that gene mutations in patients with Fanconi anemia can be corrected using NICER. FA18JTO hTERT cells, derived from a patient with Fanconi anemia and immortalized with stable hTERT expression, contained compound heterozygous mutations in the FANCA gene (c.1811delT and c.2546delC) that disrupted full-length FANCA protein expression (Fig. 8e–g). We designed a sgRNA

(sgFA18JTO_Ex20_mt20s) specifically targeting the c.1811delT mutation (Supplementary Figure 16b) and two sgRNAs targeting intron 18 or intron 20 (Fig. 8e). Following electroporation of FA18JTO hTERT cells with these sgRNAs and Nickase, gene editing was conducted under two different experimental conditions with respect to sgRNA or Nickase concentrations: a total of 3.6 pmol/μL of 3 sgRNAs + 179 ng/μL of Nickase or a total of 5.0 pmol/μL of 3 sgRNAs + 136 ng/μL of Nickase. Unexpectedly, long-range PCR demonstrated that 18.0% (16/89) and 8.9% (7/79) of the SCCs contained long deletions.

To examine whether FANCA mutations induce long deletions between tandemly generated nicks, we introduced MNs into the TK1 gene in FANCA-mutated TK6261 (FANCA$^{mt/mt}$ #C3) cells by electroporation with sgEx4_mt20s, sgS14, and Nickase, generating 94 × 3 SCCs, and subjected them to long-range PCR. The long deletion frequency in TK6261 cells was at the basal level, whereas 16.0% ± 3.8% of the FANCA$^{mt/mt}$ #C3 SCCs had long deletions (Supplementary Figure 16c−e).

We analyzed the exon 20 sequence of the FANCA gene in long-deletion-negative FA18JTO-derived SCCs using Sanger sequencing. Four of the 145 SCCs had the WT exon 20 (Fig. 8f). An immunoblotting analysis of three of these clones (FA18JTOhT-12-3, FA18 JTOhT-12-9, and FA18 JTOhT-13-40) confirmed the FANCA protein expression (Supplementary Fig. 16f). FANCD2 monoubiquitination was detected in response to mitomycin C treatment in FA18JTOhT-13-40 cells, indicating functional FANCA in this clone (Fig. 8g).

## Discussion

The use of CRISPR gene editing in embryos for therapeutic purposes remains a highly controversial topic[47]. Some have proposed that frequent IH-HR might enable germline gene therapies[48,49], whereas others have demonstrated that CRISPR-mediated gene editing via DSBs can cause chromosomal disruption, such as large deletions or rearrangements[50–52]. Consequently, the safety of IH-HR-based gene correction in zygotes has yet to be established. In addition, owing to biological and ethical concerns, the application of gene editing in human zygotes for clinical purposes is currently unfeasible. Therefore, it may be advantageous to investigate gene editing-based therapies targeting somatic cells, where IH-HR occurs less frequently.

The gene editing technique NICER is facilitated by an allele-specific primary nick and additional secondary nicks on both alleles. In a local region where a nick is present in only one allele, the allele with the nick functions as the recipient, while the other serves as the donor. This method can correct a heterozygous mutation adjacent to an allele-specific nick to match the WT sequence. MNing outperformed SNing in gene correction, achieving 10-fold greater efficiency (Figs. 1–3). The observation that SNs on both homologous chromosomes enhanced MN-IH-HR, whereas tandem nicks on one chromosome did not enhance MN-IH-HR (Fig. 3d–g), suggests that the secondary nick on the allele without the primary nick is crucial for MN-IH-HR enhancement. These results are consistent with those of previous studies on nick-induced HDR[11–16,18–24]. Although the detailed mechanisms remain unclear, the exposure of ssDNA surrounding the MNs on both homologous chromosomes and the recruitment of

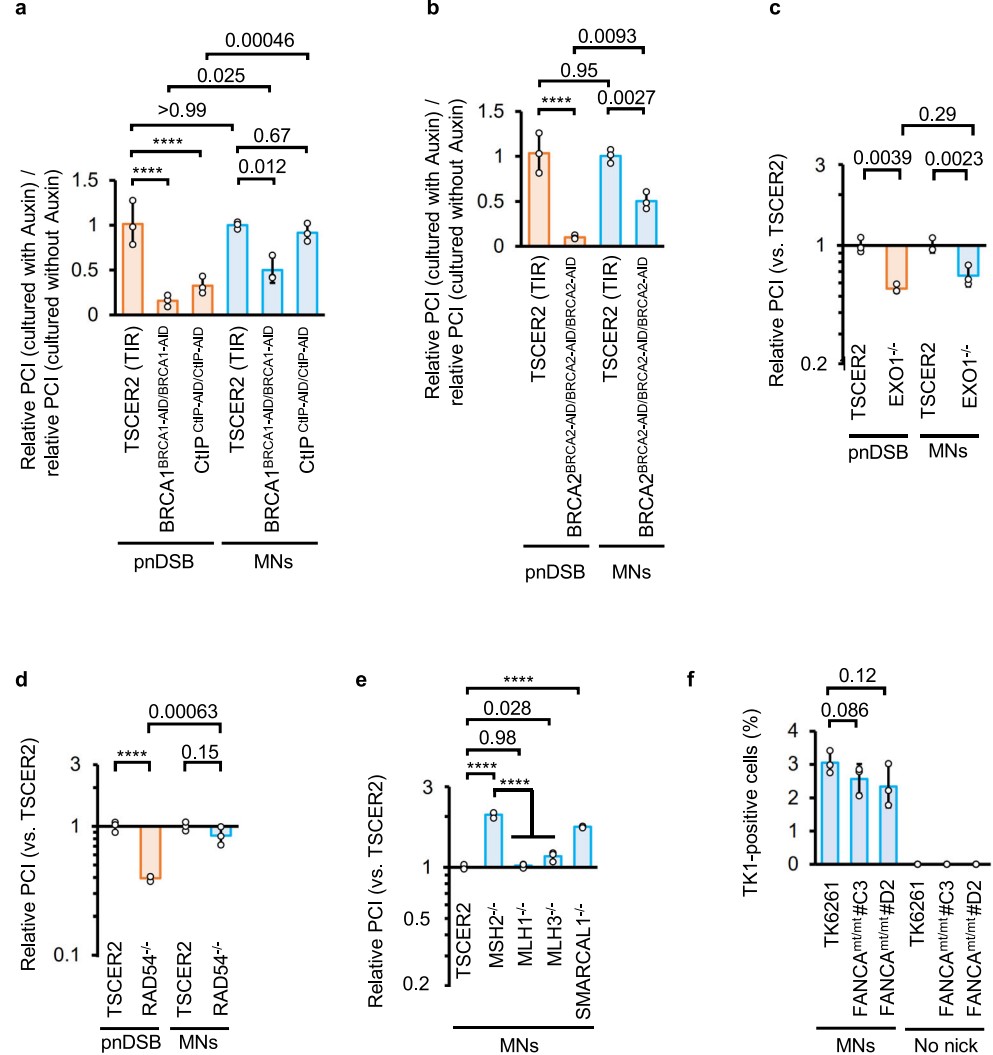

**Fig. 6 | Mechanistic insights into MN-IH-HR.** Recovery efficiency of TK1 activity in DNA repair-related gene knockout, knockdown, and mutated cells. MNs were introduced using Nickase with sgEx4_mt20s and sgS14 (light blue bars). pnDSB was introduced using Nickase with sgEx4_mt20s and sgEx4_88as (orange bars). The standard samples were TK6261_EGFP (**a**, **d**, **e**) or TK6261_mCherry (**b**, **c**) cells. **a**, **b** Ratio of the relative PCI of auxin-treated cells (1 = average PCI in TSCER2 [TIR] treated with auxin) to that of untreated cells (1 = average PCI in TSCER2 [TIR] not treated with auxin) in a proliferation assay. To deplete AID-tagged BRCA1, BRCA2, or CtIP, cells were treated with auxin starting half a day prior to electroporation, and treatment was continued for two days post-electroporation. **c–e** Relative PCIs (1 = average PCI in TSCER2) in the proliferation assay. The vertical axis represents a logarithmic scale. **f** Percentage of TK1 activity-positive cells determined by the colony formation assay. Data represent the mean ± SD from three independent experiments. Data were analyzed using a two-way ANOVA with Šídák's multiple comparisons test (**a–d**) (two-sided) or a one-way ANOVA followed by Tukey's (**e**) or Dunnett's (**f**) multiple comparisons test (two-sided). $F = 230.2$ (**e**) or 3.159 (**f**). $F_{(1, 12)} = 23.77$ (silenced gene, **a**), $F_{(2, 12)} = 39.15$ (type of DNA damage, **a**), $F_{(1, 8)} = 6.421$ (silenced gene, **b**), $F_{(1, 8)} = 95.24$ (type of DNA damage, **b**), $F_{(1, 8)} = 1.205$ (EXO1 knockout, **c**), $F_{(1, 8)} = 64.80$ (type of DNA damage, **c**), $F_{(1, 8)} = 18.13$ (RAD54 knockout, **d**), $F_{(1, 8)} = 50.69$ (type of DNA damage, **d**), $F_{(4, 10)} = 230.2$ (**e**), or $F_{(3, 9)} = 3.159$. Exact *P*-values are shown in the graphs. ****$P < 0.0001$. Source data were provided as a Source Data file.

RAD51 (Fig. 5c–e, Supplementary Fig. 17) likely contribute to increasing the occurrence of MN-IH-HR.

NICER can not only correct base substitutions and small indels but also rectify deletions larger than 600 bp, which are challenging to correct using BEs or prime editors (PEs) (Fig. 4g–i). MNing homologous chromosomes rarely results in CRISPR-dependent indels or chromosomal abnormalities (Fig. 7, Tables 1 and 2, Supplementary Fig. 15) at on- or off-target sites, and it lacks the risk of exogenous DNA random integration. Unlike BEs and PEs, NICER avoids employing deaminases or reverse transcriptases, thereby eliminating the off-target effects of these enzymes[3,53,54]. Off-target mutations caused by misrecognition of target sequences by Cas9 nickase can potentially be mitigated using high-precision nickases, such as eSpCas9(1.1)[D10A55,56] or Sniper-Cas9[D10A55,57]. The application of high-precision nickases could also improve the specific detection of heterozygous mutations,

thereby broadening the utility of NICER gene editing. Additional LOH is a potential concern; however, it can be detected through Sanger sequencing, enabling the establishment of unintended LOH-free gene-corrected cell clones.

Our ChIP-droplet digital PCR assay strongly suggested that MNing exposes long ssDNA strands on both homologous chromosomes (Fig. 5c–e). Davis et al. indicated that both DNA strands with or without nickase-generated nicks anneal to complementary ssODNs at nickase-cleaved genomic sites[14,58], supporting the idea that the length of exposed ssDNA generated by nickase is sufficient to anneal to the complementary DNA strand. Such a structure might increase the opportunities for ssDNA strands on the donor or recipient allele to anneal to each other, thereby facilitating frequent MN-IH-HRs. In contrast, when an SN is generated on one of the homologous chromosomes, ssDNA is not exposed on the other homologous

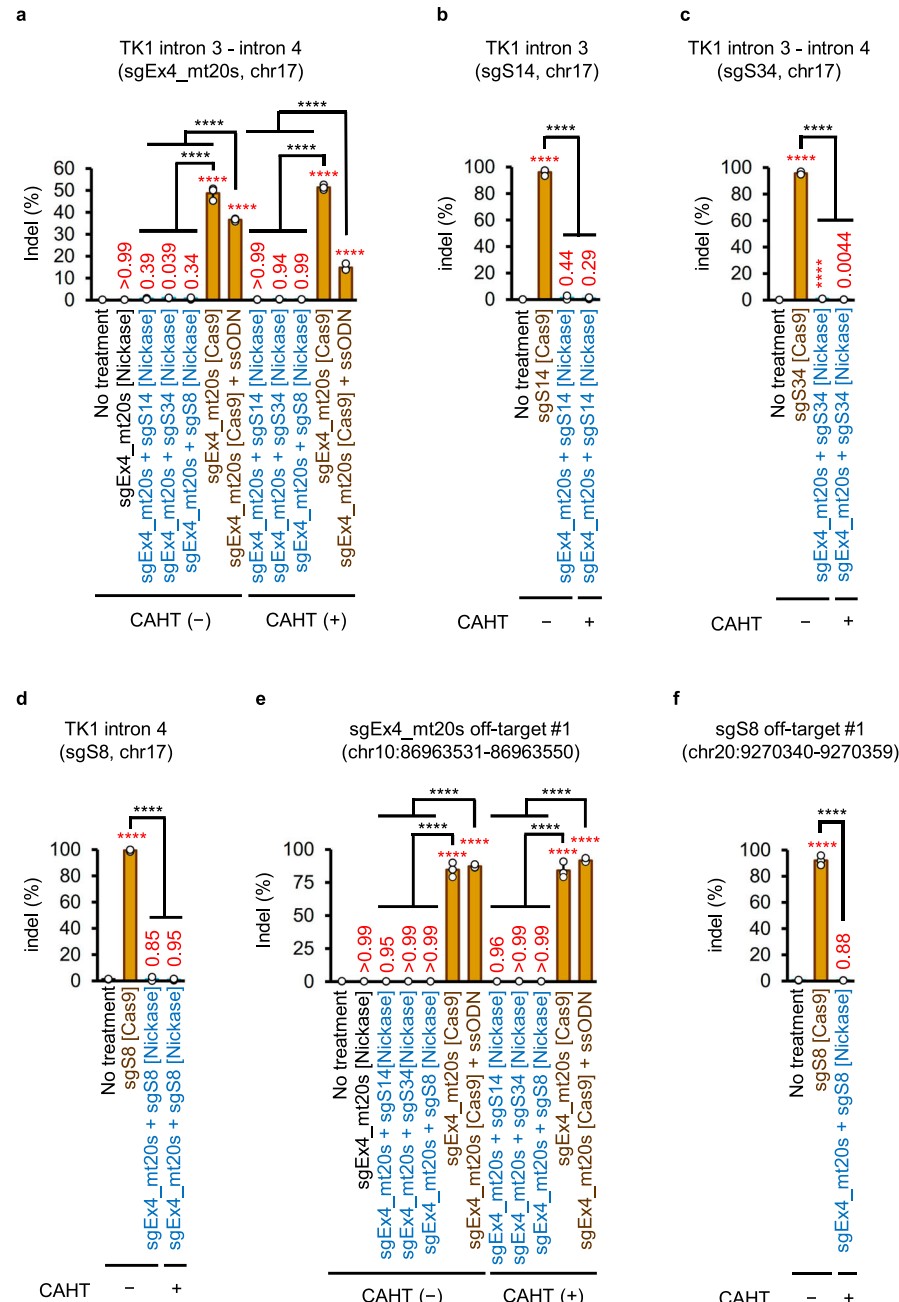

**Fig. 7 | Genomic alterations detected using AmpNGS.** Indel frequencies in regions surrounding the on-target sites (**a**–**d**) or the most probable in silico-predicted off-target sites (**e**, **f**) of the indicated sgRNAs. Genomic DNA extracted from cells electroporated with the indicated sgRNAs, and either Nickase or Cas9 mRNA was subjected to ampNGS. CHAT (−), cells cultured in SCM after gene editing; CHAT (+), TK1 activity-positive cells enriched in CHATM after gene editing; ssODN, single-strand oligodeoxynucleotide. Data represent the mean ± SD from three independent experiments. The proportion data were transformed into a continuous scale using probit transformation and subjected to a one-way ANOVA followed by Tukey's multiple comparisons test (two-sided). The *F* values were as follows: $F (11, 24) = 300.7$ (**a**), $F (3, 8) = 116.3$ (**b**), $F (3, 8) = 2,622$ (**c**), $F (3, 8) = 100.2$ (**d**), $F (11, 24) = 704.8$ (**e**), or $F (3, 8) = 520.6$ (**f**). Exact *P*-values are shown in the graphs. The red letters represent *P*-values in comparison with the data from untreated cells (no treatment). ****$P < 0.0001$. The genomic loci are denoted with GRCh38/hg38 as the reference. Source data were provided as a Source Data file.

chromosome, which may make it difficult to induce IH-HR (Supplementary Fig. 17).

Our data demonstrate that BRCA1 and BRCA2 contribute approximately half as much to MN-IH-HR as they do to pnDSB-induced IH-HR (Fig. 6a, b). This finding cannot be explained by a unilateral mechanism, suggesting that BRCA1- and BRCA2-dependent, as well as BRCA1- and BRCA2-independent pathways, are involved in MN-IH-HR (Supplementary Fig. 17). A potential pathway that relies on BRCA1 and BRCA2 is the repair of one-ended DSBs. In cycling TK6261 cells, nicks generated by Cas9 nickase may occasionally encounter the replication fork and form one-ended DSBs during replication. Consistent with this process, EXO1 knockout led to partial suppression of MN-IH-HR (Fig. 6c). One-ended DSBs can be repaired via fork reversal or remodeling, in which SMARCAL1 plays a crucial role[9]. HR or IH-HR may repair one-ended DSBs to compensate for inefficient fork reversal or remodeling in SMARCAL1$^{-/-}$ cells, potentially enhancing the efficiency of MN-IH-HR (Fig. 6e). Despite the above evidence supporting the existence of the MN-IH-HR pathway through nick-to-DSB conversion,

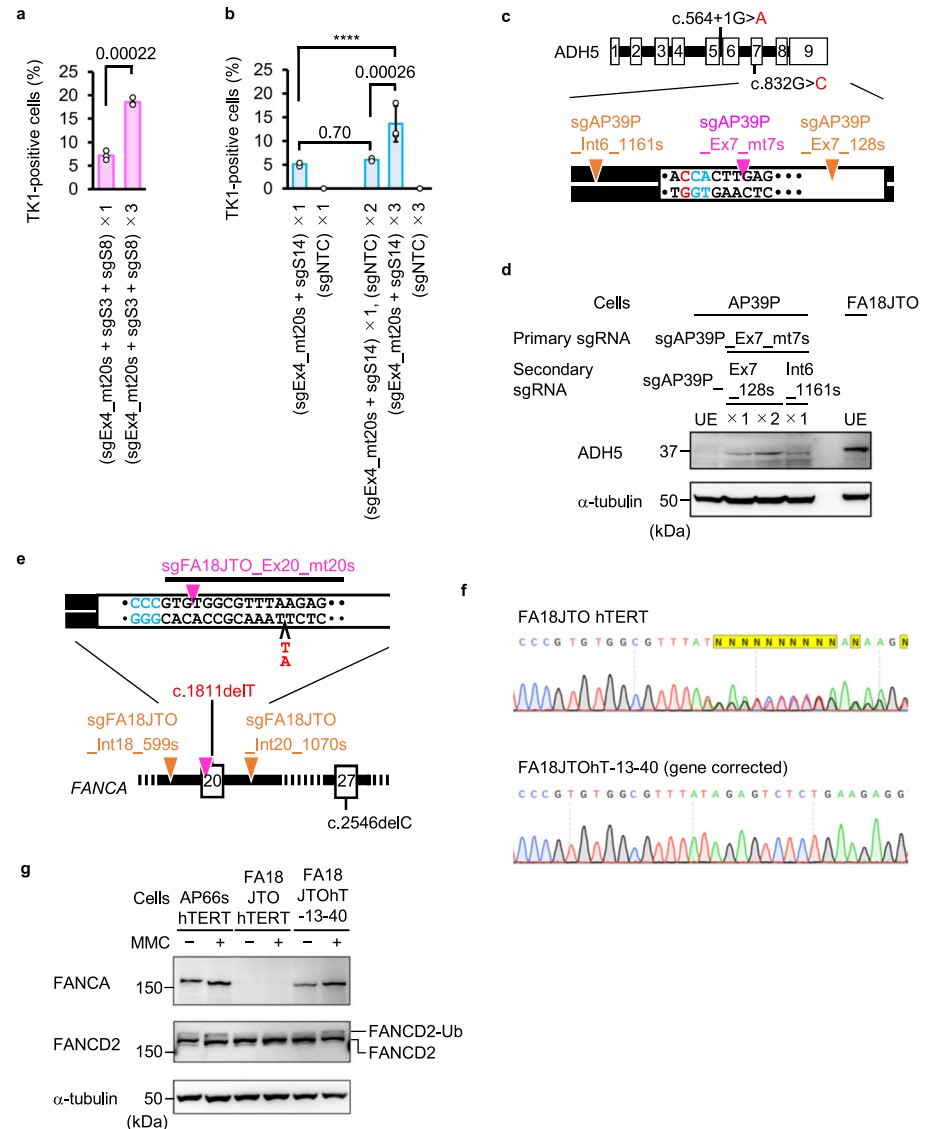

**Fig. 8 | MNs recovered the expression of gene products encoded by genes mutated in hereditary disease-derived cells. a, b** Percentage of TK1 activity-positive cells determined using a colony formation assay. The ×1, ×2, and ×3 labels indicate that cells were electroporated with the specified sgRNAs and Nickase once, twice, or three times, respectively. Data represent the mean ± SD from three independent experiments. The proportion data were adjusted using the following formula: AdX = $(X + 10^{-10})/(1 + 2 \times 10^{-10})$, where X is the original proportion and AdX is the adjusted proportion (**b**). The original proportion data (**a**) and adjusted proportion data (**b**) were transformed into a continuous scale using probit transformation and subjected to a two-tailed unpaired $t$-test (**a**, $t = 12.68$, d$f = 4$) or a one-way ANOVA followed by Tukey's multiple comparisons test (**b**, two-sided, $F$ [4, 10] = 3,676). Exact $P$-values are shown in the graphs. ****$P < 0.0001$. **c, d** The mutation in exon 7 of the ADH5 gene in AP39P cells was corrected by NICER. **c** A schematic illustration of mutations of the ADH5 gene in AP39P cells and sgRNA target sites (inverted triangles). **d** Immunoblotting for ADH5 protein in cells before (unedited [UE]) or after NICER using the specified sgRNAs. The ×1 and ×2 labels indicate that the cells underwent NICER once and twice, respectively. FA18JTO cells were used as positive controls for ADH5 protein expression. Data represent three independent experiments. **e–g** The mutation in exon 20 of the FANCA gene in FA18JTO hTERT cells was corrected by NICER. **e** A schematic illustration of mutations in the FANCA gene in FA18JTO hTERT cells and target sites of sgRNAs (inverted triangles). **f** Sanger sequencing data from FA18JTO hTERT and FA18JTOhT-13-40 cells. **g** Immunoblotting for FANCA and FANCD2 proteins in the specified cells before or after treatment with mitomycin C (MMC) is presented. AP66S hTERT cells were used as a positive control for FANCA expression. FANCD2-Ub, monoubiquitinated FANCD2. Data represent three independent experiments. Source data were provided as a Source Data file.

CtIP knockout resulted in little to no decrease in MN-IH-HR efficiency, irrespective of the location of the nick on either DNA double-helix strand (Fig. 6a and Supplementary Fig. 13e). These data suggested that the structure of one-ended DSBs arising from replication fork collisions with Cas9 nickase-generated nicks may enable CtIP-independent IH-HR.

BRCA1 and BRCA2 are essential factors for HR, leading to the understanding that the MN-IH-HR pathway, independent of BRCA1 and BRCA2, is not involved in nick-to-DSB conversion. This idea is supported by previous studies that have revealed the presence of BRCA2- and RAD51-independent nick-induced HDR[58]. In fact, we observed correction of the TK1 gene by NICER, even in cells arrested in the G1 phase, to prevent replication. These findings suggest that nicks generated by nickase in the G1 phase are directly repaired by the HR-like DNA repair pathway. If a nick adjacent to the mutation is repaired by MN-IH-HR without nick-to-DSB conversion, the mutation may be corrected only in the damaged strand, resulting in a mismatch in the DNA helix (Supplementary Fig. 17). Indeed, our data indicate the involvement of mismatch generation during the MN-IH-HR process in at least a portion of the MN-introduced cell populations

(Fig. 6e, Supplementary Fig. 18). However, a comprehensive and detailed experimental validation using cells with suppressed replication is required to elucidate the molecular mechanism underlying MN-IH-HR without nick-to-DSB conversion. From a clinical perspective, it would be intriguing to investigate whether or not NICER can correct heterozygous mutations in differentiated non-dividing neurons or muscle cells.

In conclusion, our findings demonstrate that MNing homologous chromosomes induce MN-IH-HR and that the application of NICER enables scarless gene correction of heterozygous mutations. This study provides insight into genome editing technologies and broadens the potential clinical applications of CRISPR/Cas9-mediated gene editing.

## Methods

### sgRNAs, gRNA, tracrRNA, Cas9 mRNA, Cas$^{D10A}$ mRNA, Cas9$^{H840A}$ mRNA, and Cas9 protein

Customized Alt-R CRISPR-Cas9 sgRNAs (mN*mN*mN*rNrNrNrNr NrNrNrNrNrNrNrNrNrNrNrNrNrNrNrGrUrUrUrUrArGrArGrCrUrArGrArAr-ArUrArGrCrArArGrUrUrArArArArArUrArArGrGrCrUrArGrUrCrCrGrUrUr ArUrCrArArCrUrUrGrArArArArArArGrUrGrGrCrArCrCrGrArGrUrCrGrGr UrGrCmU*mU*mU*rU, mN: 2' O-methyl RNA base, rN: RNA base, *: phosphorothioate bond), Alt-R CRISPR-Cas9 tracrRNA (rArGrCrAr UrArGrCrArArGrUrUrArArArArUrArArGrGrCrUrArGrUrCrCrGrUrUrAr UrCrArArCrUrUrGrArArArArArGrUrGrGrCrArCrCrGrArGrUrCrGrGrUr GrCrUrUrU, rN: RNA base, #1072532) and customized Alt-R CRISPR-

Cas9 gRNA were purchased from IDT. The target sequences of sgRNAs and gRNAs are listed in Supplementary Data 1. 5-methoxyuridine CleanCap Cas9 mRNA (L-7206) and 5-methoxyuridine CleanCap Cas9$^{D10A}$ mRNA (L-7606) were purchased from TriLink Biotechnologies. 5-methoxyuridine CleanCap Cas9$^{H840A}$ mRNA was synthesized using TriLink Biotechnologies.

### Cells and cell culture

TK6(IVGT) (JCRB1435), FA18JTO hTERT (JCRB3007), FA18JTO (JCRB0315), AP39P (JCRB3068), and AP66s hTERT (JCRB3066) were obtained from the Japanese Collection of Research Bioresources Cell Bank (Osaka, Japan). TSCER2 and BLM-knockout TSCER2 cells[27] were generously provided by Dr. Honma (available from Dr. M. Yasui, National Institute of Health Sciences, Japan, upon finalizing an MTA). Other cells derived from TSCER2 were established by H. Sasanuma (available from H. Sasanuma, Tokyo Metropolitan Institute of Medical Science, Japan, upon finalizing an MTA). TK6261 and cells derived from TK6261 were established by S. Nakada (available from S. Nakada, Osaka University, Japan, upon finalizing an MTA). TK6(IVGT) cells, TSCER2 cells, TK6261 cells, or cells derived from these cells were cultured in SCM: RPMI1640 (Nacalai Tesque) supplemented with 5% horse serum (Invitrogen), 0.1 mg/mL sodium pyruvate (Nacalai Tesque), and 100 U/mL penicillin and l00 μg/mL streptomycin (Nacalai Tesque) at 37 °C with 5% $CO_2$. After electroporation, TK6(IVGT) cells, TSCER2 cells, TK6261 cells, or cells derived from these cells were cultured in the recovery medium RPMI1640 (Nacalai Tesque) supplemented with 10%

## Table 1 | Numbers of SNVs and indels evaluated by a GATK analysis of WGS of SCCs established before or after gene correction

| Gene editing | | Number of analyzed off-targets | Number of analyzed clones | Number of SNVs and indels (×10$^{-2}$/off-target/clone) at regions surrounding the predicted off-target sites | | | |
|---|---|---|---|---|---|---|---|
| | | | | ±10 bp | ±25 bp | ±50 bp | ±100 bp |
| sgEx4_mt20s + sgS14 [Nickase] | No treatment | 198 | 12 | 0.04 | 0.04 | 0.04 | 0.17 |
| | Gene corrected | 198 | 13 | 0.00 | 0.00 | 0.00 | 0.04 |
| sgEx4_mt20s + sgS8 [Nickase] | No treatment | 198 | 12 | 0.00 | 0.00 | 0.00 | 0.04 |
| | Gene corrected | 198 | 10 | 0.00 | 0.00 | 0.00 | 0.20 |
| sgEx4_mt20s + sgS34 [Nickase] | No treatment | 198 | 12 | 0.04 | 0.04 | 0.21 | 0.75 |
| | Gene corrected | 198 | 9 | 0.00 | 0.00 | 0.00 | 0.22 |
| sgEx4_mt20s [Cas9] + ssODN | No treatment | 99 | 12 | 0.00 | 0.00 | 0.00 | 0.00 |
| | Gene corrected | 99 | 15 | 2.53 | 3.00 | 3.00 | 3.60 |

The number of SNVs and indels (×10$^{-2}$/off-target/clone) in the ±10, ±25, ±50, and ±100-bp regions surrounding the 3' end of the off-target sequences was analyzed using the GATK method. Regions surrounding the 99 off-target sites of each sgRNA predicted in silico were analyzed. SNV, single nucleotide variant; WGS, whole Genome Sequencing; SCC, single-cell derived clone.

## Table 2 | Numbers of SVs and long indels evaluated by a MANTA analysis of WGS of SCCs established before or after gene correction

| Gene editing | | Number of analyzed off-targets | Number of analyzed clones | Number of SVs and long indels (×10$^{-3}$/off-target/clone) at regions surrounding the predicted off-target sites | | | |
|---|---|---|---|---|---|---|---|
| | | | | ±10 bp | ±25 bp | ±50 bp | ±100 bp |
| sgEx4_mt20s + sgS14 [Nickase] | No treatment | 198 | 12 | 0.00 | 0.00 | 0.00 | 0.00 |
| | Gene corrected | 198 | 13 | 0.00 | 0.00 | 0.00 | 0.00 |
| sgEx4_mt20s + sgS8 [Nickase] | No treatment | 198 | 12 | 0.00 | 0.00 | 0.00 | 0.00 |
| | Gene corrected | 198 | 10 | 0.00 | 0.00 | 0.00 | 0.00 |
| sgEx4_mt20s + sgS34 [Nickase] | No treatment | 198 | 12 | 0.00 | 0.00 | 0.00 | 0.00 |
| | Gene corrected | 198 | 9 | 0.00 | 0.56 | 0.56 | 0.56 |
| sgEx4_mt20s [Cas9] + ssODN | No treatment | 99 | 12 | 0.00 | 0.00 | 0.00 | 0.00 |
| | Gene corrected | 99 | 15 | 2.00 | 3.33 | 3.33 | 4.00 |

The number of SVs and indels (×10$^{-3}$/off-target/clone) in the ±10, ±25, ±50, and ±100-bp regions surrounding the 3' end of the off-target sequences was analyzed using the MANTA method. Regions surrounding the 99 off-target sites of each sgRNA predicted in silico were analyzed. SV, structural variant; WGS, whole Genome Sequencing; SCC, single-cell derived clone.

horse serum (Invitrogen) and 0.2 mg/mL sodium pyruvate (Nacalai Tesque) at 37 °C with 5% $CO_2$ for 1 day. AP39P cells were cultured in RPMI1640 medium with 20% fetal bovine serum at 37 °C and 5% $CO_2$. FA18JTO, FA18JTO hTERT, and AP66s hTERT cells were cultured in MEM alpha with 10% heat-inactivated fetal bovine serum at 37 °C with 5% $CO_2$. No cell lines featured in the database of commonly mis-identified lines were used in this study. The cells used in this study are listed in Supplementary Data 6.

### Generation of TK6261 cells
TK6(IVGT) cells ($100 \times 10^4$ cells in 10 μL of buffer R) were mixed with 10 μL of ribonucleoprotein (RNP complex, prepared according to the manufacturer's protocol [IDT]; 52.8 pmol of crRNA, 52.8 pmol of tracrRNA [IDT], and 8 μg of Alt-R S.p.Cas9 nuclease V3 [IDT] in buffer R [Invitrogen]). The mixture (10 μL) was electroporated using a Neon Transfection System with a pulse voltage of 1500 V, pulse width of 10 ms, and pulse number of 3. An SCC of TK626 harboring an 8-bp deletion in exon 5 of the TK1 gene was established by limited dilution. TK626 cells were seeded onto 96-well plates at 0.5/well, and an SCC of TK6261 cells was established.

### Generation of TK6261_S43v11, TK6261_S43v44, TK6261_S62v49, and TK6261_int5v32 cells
TK6261 cells ($50 \times 10^4$ cells in 10 μL of buffer R) were electroporated (pulse voltage, 1500 V; pulse width, 10 ms; pulse number, 3) with 2 μg of pCMV_BE4max_P2A_GFP[59] (Addgene #112099) and cultured in the recovery medium for 1 day. Cells ($50 \times 10^4$ in 10 μL of buffer R [Invitrogen]) were electroporated with 0.32 μL of 100 pmol/μL sgS43 (for TK6261_S43v11 and TK6261_S43v44), sgS62 (or TK6261_S62v49), or sgEx6_-92s (or TK6261_int5v32) and then cultured for a day in recovery medium. GFP-positive cells were sorted into 96-well plates at 1/well using an Aria III cell sorter (BD) in single-cell mode and cultured in the SCM. We analyzed 94 SCCs using Sanger sequencing, confirming the creation of heterozygous nucleotide substitutions at the target sites. Candidate clones were subjected to long-range PCR, TOPO cloning, and Sanger sequencing to confirm the presence of alleles with nucleotide substitutions. Finally, TK6261_S43v11, TK6261_S43v44, TK6261_S62v49, and TK6261_int5v32 cells were established.

### Generation of TK6261_EGFP and TK6261_mCherry cells
The hROSA26 CMV-MCS-Hygro (pMK247) plasmid[60] was modified to target the ROSA26 locus. The HYGROMYCIN[R] gene in the pMK247 plasmid was replaced with the PUROMYCIN[R] gene (hROSA26 CMV-MCS-Puro) using the GeneArt Seamless Cloning Enzyme Mix (Thermo Fisher). The EGFP and mCherry genes were cloned into hROSA26 CMV-MCS-Puro (hROSA26 CMV-EGFP-Puro or hROSA26 CMV-mCherry-Puro) (GeneScript). TK6261 cells were electroporated using hROSA26 CRISPR-pX330 (Addgene #105927)[60] and either hROSA26 CMV-EGFP-Puro or hROSA26 CMV-mCherry-Puro. EGFP- or mCherry-positive cells were sorted using an Aria III cell sorter in single-cell mode and cultured in SCM containing 0.5 μg/ml puromycin.

### Generation of FANCA-mutated TK6261 cells
To disrupt FANCA protein expression, we introduced frameshift deletions in exon 27 of both alleles using CRISPR/Cas9 into TK6261 cells. TK6261 cells ($70 \times 10^4$ cells in 10 μL of buffer R [Invitrogen]) were mixed with 0.9 μL of 100 pmol/μL sgFANCA_exon 27_45s (IDT), 0.9 μL of 1 μg/μL 5-methoxyuridine CleanCap Cas9 mRNA (TriLink Biotechnologies), and 2.2 μL of R buffer (Invitrogen). The mixture (10 μL) was then subjected to electroporation using a Neon Transfection System (Invitrogen) with the following parameters: pulse voltage, 1500 V; pulse width, 10 ms; pulse number, 3. The cells were cultured in the recovery medium for 1-day post-electroporation before being transferred to the SCM. SCCs of TK6261

harboring biallelic mutations in exon 27 of FANCA were established using limited dilution.

### Generation of BRCA2[BRCA2-AID/BRCA2-AID/BRCA2-AID] and POLQ[−/−] and cells
To construct targeting vectors for the BRCA2[BRCA2-AID/BRCA2-AID/BRCA2-AID] cells (BRCA2 genes are located on trisomic chromosome 13 in TSCER2 cells), the left and right arms were amplified using specific primers, as follows: left_forward, 5′-ACTGTGGCATAGAGAGAGGTTAAGC; left_reverse, 5′-GATATATTTTTTAGTTGTAATTGTG; right_forward, 5′-TGTTGCACAATGAGAAAAGAAATTA; and right_reverse, 5′-CATGGTAAAACCCAGTCTCTACTAA. The left and right arms were assembled with pBS-mAID-GFP-loxP-MARKER[R] (NEOMYCIN[R], HYGROMYCIN[R], and HISTIDINOL[R]) digested with EcoNI and SmaI using the GeneArt Seamless Cloning Enzyme Mix (Thermo Fisher)[34]. The gRNA (5′-ACCTTTCCAGTTTATAAGAC) was inserted into the BbsI site of the pX330 vector (Cat# 42230, Addgene). The 3 resulting targeting vectors and the pX330-gRNA vector were transfected into WT cells expressing TIR1[34], and the transfectants were incubated with neomycin-, hygromycin-, and histidinol-containing medium for 10 days.

To generate targeting vectors for POLQ[−/−] cells, the left and right arms were amplified using the following specific primers: left_forward, ATTTCTCCACTAGTTCCAGGCTTCT; left_reverse, TAAAACGTTAATTCATTGGCTAAGTGGC; right_forward, GGCCAAGAATTCAAGTGTTGATCAAGG; and right_reverse, TGCCCAGCCTATTACCTTGTTTTATACATG. The arms were assembled with DT-ApA/NEO[R] and DT-ApA/Hygro[R] vectors using the GeneArt Seamless Cloning Enzyme Mix. The gRNA (5′-AGTTCAGATGACATCGCTGA or AGTCGCACACTGCTACAGGTG) was inserted into the BbsI site of the pX330 vector. The 4 resulting vectors were transfected into TSCER2 cells, and the transfectants were incubated with neomycin- and hygromycin-containing medium for 10 days.

### Electroporation for a gene-editing assay
For TK6261 cells, TSCER2 cells, or their derived cells, $70 \times 10^4$ cells in 10 μL of buffer R (Invitrogen) were mixed with 0.9 μL of 100 pmol/μL sgRNAs (IDT), 0.9 μL of 1 μg/μL 5-methoxyuridine CleanCap Cas9, Cas9[D10A] or Cas9[H840A] mRNA (TriLink Biotechnologies), and 2.2 μL of R buffer (Invitrogen). The mixture (10 μL) was then subjected to electroporation using a Neon Transfection System (Invitrogen) with the following parameters: pulse voltage, 1500 V; pulse width, 10 ms; pulse number, 3. The cells were cultured in the recovery medium for 1-day post-electroporation before being transferred to the SCM.

For FA18JTO hTERT cells, $21 \times 10^4$ cells in 10 μL of R buffer were mixed with 1.5 or 2.1 μL of 100 pmol/μL sgRNAs and 2.5 or 1.9 μL of 1 μg/μL 5-methoxyuridine Nickase. The mixture (10 μL) was subjected to electroporation using a Neon Transfection System (Invitrogen) with the following parameters: pulse voltage, 1500 V; pulse width, 20 ms; pulse number, 2.

For AP39P cells, $28 \times 10^4$ cells in 10 μL of solution R were mixed with 0.9 μL of 100 pmol/μL sgRNAs and 0.9 μL of 1 μg/μL 5-methoxyuridine Cas9[D10A] mRNA. The mixture (10 μL) was subjected to electroporation using a Neon Transfection System (Invitrogen) with the following parameters: pulse voltage, 1050 V; pulse width, 30 ms; pulse number, 2.

### Colony formation assays
To assess colony formation, we followed the procedure described by Nakajima et al.[19] In brief, cells were seeded in 200 μL of CHATM (SCM with 10 μM 2′-deoxycytidine [Sigma-Aldrich], 200 μM hypoxanthine [Sigma-Aldrich], 100 nM aminopterin [Sigma-Aldrich], and 17.5 μM thymidine [Sigma-Aldrich]) at 1, 5, 10, 15, 20, 30, 100, or 200 cells/well in two 96-well plates. To analyze the plating efficiency, cells were seeded in 200 μL of SCM at 1 cell/well in two 96-well plates. Two weeks later, the colony-positive wells were analyzed. The percentage of

aminopterin-resistant (TK1 activity-positive) cells was calculated using the following equation:

$$\frac{PC/SC/192}{PS/192} \times 100, \qquad (1)$$

where PC is the number of colony-positive wells in CHATM, SC is the number of seeded cells in a well in CHATM, and PS is the number of colony-positive wells in SCM.

## Proliferation assay

For the assay shown in Fig. 2a, we electroporated mCherry-expressing cells (TK6261-mCherry cells) with Nickase, sgEx4_mt20s, and sgS0 as standard samples. In addition, we electroporated EGFP-expressing cells (TK6261-EGFP cells) with Nickase, sgEx4_mt20s, and 1 of the 67 sgRNAs targeting various 20 bp DNA sequences located approximately 8.2 kb upstream to 8.6 kb downstream of the sgEx4_mt20s targeting sequence in the TK1 gene as test samples (Supplementary Fig. 1b, left panel). The cells were cultured for one day in a recovery medium and for three days in SCM. Subsequently, TK6261−mCherry and TK6261−EGFP cells were mixed at a 1:1 cell-count ratio. Some of the mixed cells were cultured in SCM for seven or eight days, whereas the rest were cultured in CHATM for four days before being cultured in SCM for an additional 3 or 4 days. After culturing, the percentages of TK6261−mCherry and TK6261−EGFP cells were measured using a FACS Aria III (BD) or Fusion (BD) cell sorter using the FACS Diva software program (version 8.0.1; BD). The PCI was calculated using the following equation:

$$PCI = \frac{GC \times CS}{GS \times CC}, \qquad (2)$$

where GC is the percentage of EGFP-positive cells in CHAT-treated cells, GS is the percentage of EGFP-positive cells in SCM, CC is the percentage of mCherry-positive cells in CHAT-treated cells, and CS is the percentage of mCherry-positive cells in SCM.

In the alternate proliferation assay, we electroporated either TK6261-mCherry or TK6261-EGFP cells with Nickase and sgRNAs as standard samples. In addition, we electroporated fluorescence-negative or fluorescence-weak cells using Nickase and sgRNAs as test samples. Following electroporation, the cells were cultured in the recovery medium for one day and then in SCM for 3 days. TK6261−mCherry/-EGFP cells and fluorescence-negative/-weak cells were mixed in a 1:1 cell-count ratio. Some of the mixed cells were cultured in SCM for seven or eight days, whereas the rest were cultured in CHATM for four days before being cultured in SCM for three or four days. After culturing, the percentage of fluorescent-positive (or highly fluorescent) or fluorescence-negative (or weakly fluorescent) cells was determined using a FACS Aria III (BD) or Fusion (BD) cell sorter using the FACS Diva software program (version 8.0.1; BD). The PCI was calculated using the following equation:

$$PCI = \frac{fNC \times (1 - fNS)}{fNS \times (1 - fNC)}, \qquad (3)$$

where fNC represents the percentage of fluorescence-negative (or weakly fluorescent) cells among cells cultured in CHATM, and fNS represents the proportion of fluorescence-negative (or weakly fluorescent) cells among cells cultured in SCM.

## Auxin degron system-mediated AID-tagged protein knockdown

BRCA1[BRCA1-AID/BRCA1-AID], BRCA2[BRCA2-AID/BRCA2-AID/BRCA2-AID], and CtIP[CtIP-AID/CtIP-AID] cells were cultured with 250 μM auxin from 0.5 days prior to electroporation until 2.0 days after to deplete BRCA1-AID, BRCA2-AID, or CtIP-AID. Since prolonged depletion of these proteins significantly

suppressed proliferation, the cells were treated with auxin during this period. Because of the leaky degradation of AID-tagged proteins in the AID system, the expression of BRCA1-AID in BRCA1[BRCA1-AID/BRCA1-AID] cells, BRCA2-AID in BRCA2[BRCA2-AID/BRCA2-AID/BRCA2-AID] cells, or CtIP-AID in CtIP[CtIP-AID/CtIP-AID] cells cultured in SCM without auxin was considerably lower than that of BRCA1, BRCA2, and CtIP in TSCER2(TIR) cells, respectively. Nevertheless, the finding of a reduced expression of BRCA1-AID, BRCA2-AID, and CtIP-AID was sufficient to support the IH-HR of pnDSBs. Of note, further degradation of BRCA1-AID, BRCA2-AID, and CtIP-AID by auxin degrons almost completely inhibited the IH-HR of pnDSBs (Fig. 6a, b). Consequently, it was deemed appropriate to use these cells to validate the dependency of MN-IH-HR on BRCA1, BRCA2, and CtIP.

## G1 arrest caused by palbociclib

The cells were treated with 4 μM palbociclib (Sigma-Aldrich) 24 h prior to electroporation. The electroporated cells were further incubated in the recovery medium with or without 4 μM palbociclib for 3 days and then washed twice with SCM and cultured in SCM for 4 days, after which the cells were subjected to a proliferation assay. For the cell cycle analysis, cells were fixed with paraformaldehyde, treated with 200 μg/mL RNase A (Invitrogen), stained with 20 μg/mL propidium iodide (Nacalai Tesque), and analyzed with a FACS Aria III (BD) cell sorter using the FACS Diva software program (version 8.0.1; BD).

## SCC generation and genomic DNA extraction

Cells were sorted into 96-well plates at 1/well using an Aria II or Aria III cell sorter (BD) in single-cell mode and cultured in the SCM. After five to seven days, genomic DNA was extracted using a Kaneka Easy DNA Extraction Kit version 2 (Kaneka). The cell suspensions (20–30 μL) were transferred to a 96-well V-bottom plate. The cell suspension was then centrifuged, and the cell culture medium was removed from the 96-well V-bottom plate. The cells were lysed in 20 μL of solution A (Kaneka) and incubated at 98 °C for 10 min; after that, 2.8 μL of solution B (Kaneka) was added to the cell lysate. For WGS or AmpNGS, genomic DNA was extracted from SCCs or cell populations using a MonoFas gDNA Cultured Cell Extraction Kit VI (Animos), following the manufacturer's protocol.

## Short- or long-range PCR

For short-range (<1 kb) PCR, 1 μL of genomic DNA extracted using the Kaneka kit was mixed with KOD Plus Neo Kit reagent (TOYOBO) following the manufacturer's protocol in a 20-μL reaction volume and subjected to two-step PCR. For long-range (>1 kb) PCR, genomic DNA extracted using the Kaneka kit was diluted to 1:10–1:200, whereas genomic DNA extracted using the MonoFas kit was diluted to 2 ng/μL in TE buffer. Subsequently, 1 μL of the diluted DNA was mixed with KOD FX Neo Kit reagent (TOYOBO) following the manufacturer's protocol in a 20-μL reaction volume and subjected to two-step PCR. The primers used are listed in Supplementary Data 7.

After PCR, the reaction mixture (1 μL), Gel Loading Dye Purple (×6) (NEB, 1 μL), and ddH2O (4 μL) were combined and electrophoresed on either 0.9% (TAE) or 1.2% (TBE) agarose gel using TAE or TBE buffer, respectively. The gels were then stained with Ultrapower DNA SafeDye (Gellex) or GelGeen Nucleic Acid Stain (Biotinum) and illuminated with blue LED light (LED-100; AMZ System Science) for visualization. The images were captured using a Canon PowerShot G1X camera). The image was tonally inverted and autocontrast-corrected using the Photoshop software program (version 24.6; Adobe), with sections cropped as necessary.

## Sanger sequencing

The PCR products were purified using an ExoSAP-IT Kit (Thermo Fisher Scientific) or a NucleoSpin Gel and PCR Clean-up Kit (Macherey-Nagel) and then mixed with sequencing primers for subsequent Sanger

sequencing performed by Eurofins Genomics. Data were analyzed using Snapgene (6.2; GSL Biotech). The primers used are listed in Supplementary Data 7.

## Immunoblotting

Cells were directly lysed in sodium dodecyl sulfate (SDS) sample buffer, and the lysates were boiled at 100 °C for 5 min. Total cell lysates were separated on e-PAGEL gels (Atto) by SDS-polyacrylamide gel electrophoresis (PAGE) and transferred to nitrocellulose membranes using an iBlot Gel Transfer System (Invitrogen). The membranes were blocked with 5% skim milk (Nacalai Tesque) in Tris-buffered saline (25 mmol/L-Tris, 137 mmol/L-NaCl, 2.68 mmol/L-KCl) with 0.1 v/v %-Polyoxyethylene Sorbitan Monolaurate (TBST; Nacalai Tesque) (SM/TBST), stained with primary antibodies diluted in 5% bovine serum albumin (BSA; Sigma-Aldrich) in TBST (BSA/TBST) or SM/TBST overnight, washed with TBST, incubated with horseradish peroxidase-conjugated secondary antibodies (Promega) diluted in SM/TBST, and washed with TBST. The Western Lightning ECL Pro Reagent Kit or Western Lightning ECL Ultra Reagent Kit (Perkin Elmer) was used for the chemiluminescence analysis. Chemiluminescent signals were detected using a LAS4000 mini instrument (GE Healthcare) and the ImageQuant LAS4000 software program (version 1.3; GE Healthcare). The acquired images were cropped using Photoshop (version 24.6; Adobe). The antibodies used were as follows: Anti-TK1 (CST; 8960 S, 1:1000 in BSA/TBST), anti-FANCA (CST; 14657 S,1:500 in BSA/TBST), Anti-FANCD2 (Novus; NB100-182, 1:2000 in SM/TBST), anti-tubulin (Sigma-Aldrich; T6074, 1:2,000 in SM/TBST), Anti-BRCA1 (Santa Cruz; D-9, sc-6954 1:100 in SM/TBST), Anti-BRCA2 (Millipore; OP-95, Ab-1, 1:1000 in SMTBST), anti-CtIP (Bethyl Laboratories; A300-488A, 1:500 in SM/TBST), Anti-ADH5 (abcam; ab174283,1:1000 in SM/TBST), Anti-ADH5 (Proteintech; 11051-1-AP, 1:500 in SM/TBST), Anti-Rabbit IgG HRP-conjugate (Promega; W401B, 1:2000 in SM/TBST), and Anti-Mouse IgG HRP-conjugate (Promega; W402B, 1:2,000 in SM/TBST). The antibodies used are listed in Supplementary Data 8. See the Source Data file and/or Supplementary Figs. 19–23 for the full scan blots.

## ChIP-droplet digital PCR assays

Cells were crosslinked for 1 h in 0.5 mg/ml N,N′-disuccinimidyl glutarate, and for 20 min in 1% formaldehyde. A total of $1 \times 10^7$ cells were suspended in 130 μL of ChIP buffer (0.25% SDS, 1% Triton X-100, 150 mM NaCl, 1 mM EDTA, 0.5 mM EGTA, 20 mM HEPES) and sonicated using an M220 system (Covaris). The supernatant (60 μL) was collected and immunoprecipitated with 10 μL of Rad51 (70-002, lot 1; BioAcademia) or 3 μg of RPA (MABE285; Sigma–Aldrich) antibodies in ChIP buffer containing 0.1% BSA and a protease inhibitor. The samples were incubated with IgG/A-conjugated magnetic beads.

ChIP samples were treated with 60 μL buffer containing 200 mM NaCl and 0.1 mg/ml proteinase K for 4 h at 65 °C for decrosslinking and protease treatment. The samples were column-purified using MinElute columns and eluted with 25 μL of buffer E (Qiagen). ChIP samples (1 μL) were subjected to droplet digital PCR (total 20 μL) using ddPCR Eva-Green Supermix (Bio-Rad). The PCR primers used are listed in Supplementary Data 7. The PCR conditions were as follows: 95 °C for 5 min, 95 °C for 30 s, 58 °C for 1 min, for 45 cycles. The amplified samples were analyzed using the QX100 Droplet Digital PCR system (Bio-Rad) with the QuantaSoft software program (version 1.7; Bio-Rad). The antibodies used are listed in Supplementary Data 8.

## Cas9-gRNA on-target score and in silico-predicted off-target sites

On-target scores and off-target sites (GRCh38/hg38) were determined using a support vector machine built from a large experimental dataset generated by IDT. This information regarding on-target scores and predicted off-target sites is available on the IDT website at https://sg.idtdna.com/site/order/designtool/index/CRISPR_SEQUENCE.

## Amplicon-based next-generation sequencing (AmpNGS)

To enrich TK1 activity-positive cells, >$10^7$ cells were cultured in CHATM for 1 week and then in SCM for 3 or 4 days. Genomic DNA was subsequently extracted using the MonoFas gDNA Cultured Cell Extraction Kit VI (Animos). AmpNGS was performed by Fasmac Inc. KOD Plus Neo was used for PCR; 1–5 ng of gDNA was subjected to the first round of PCR using region-specific primers in a 20-μL volume. All forward primers were labeled with adapter 5′-ACACTCTTTCCCTACACGACGCTCTTCCGATCT-, and all reverse primers were labeled with adapter 5′-GTGACTGGAGTTCAGACGTGTGCTCTTCCGATCT-. The sequences of the first PCR primers, excluding adapter sequences, are listed in Supplementary Data 7. PCR products were purified using AMpure XP (Beckman Coulter) or a NucleoSpin Gel and PCR Clean-up Kit (Macherey-Nagel). Subsequently, the purified PCR products were subjected to a second round of PCR using forward primers (AATGATACGGCGACCACCGAGATCTACAC-[Index]-ACACTCTTTCCCTACACGACGC) and secondary primers (CAAGCAGAAGACGGCATACGAGAT-[Index]-GTGACTGGAGTTCAGACGTGTG). The secondary PCR products were purified using AMpure XP (Beckman Coulter), diluted to 2 nM, and run on a MiSeq System using a MiSeq Reagent Kit v2 or v3 (500 or 600 cycles; Illumina). Raw reads were filtered (quality filtering using sickle [version 1.33; najoshi], sequence trimming using the Fastx toolkit [version 0.0.14; Ohio Supercomputer Center], and merging of overlapping paired-end reads using FLASH [version 2.2.00; Johns Hopkins University Center for Computational Biology]). At least 30,000 (Fig. 2b) or 50,000 (all other Figures) reads were analyzed and visualized using CrispR Variants (version 1.24.0; Bioconductor) and the R software program (version 4.3.1; The R Development Core Team). The sequence at the region ±25 bp from the predicted cleavage site is shown in Fig. 2b, whereas the sequence at the region 67 bp (5′ direction) + 56 bp (3′ direction) surrounding the predicted cleavage site was analyzed in all Figures except Fig. 2b. A portion of the analyzed results presented in Fig. 2b, Fig. 7a, Supplementary Fig. 3c, and Supplementary Fig. 11c were derived from the same raw data. A portion of the analyzed results shown in Fig. 4b and Supplementary Fig. 4e originated from the same raw data. Data used in each figure are listed in Supplementary Fig. 9.

## Detection of off-target variants in genome-edited cells by WGS

SCCs were established using individual genome-edited cell lines. Genomic DNA was isolated using a MonoFas gDNA Cultured Cell Extraction Kit VI (Animos). Next-generation WGS was performed in-house using the MGI sequencers DNBseq T7 and G400 with paired-end (PE) flow cells to obtain 150-bp PE reads with 30× coverage.

The WGS data were analyzed using our standard WGS pipeline[61,62]. In brief, low-quality reads, and sequencing adapters were trimmed and quality-checked using the fastp program[63] (version 0.20.1). The trimmed reads were mapped to the human reference genome (GRCh37/hg19) using BWA[64] (version 0.7.17-r1188). Duplicate reads were removed using biobambam2 software program[65] (version 2-2.0.72). The reads were processed using the IndelRealigner and BaseRecalibrator programs in the Genome Analysis Toolkit (GATK Queue-3.5) software program[66]. To identify SVs, base-recalibrated reads were processed using the Illumina Manta software program[67] (version 1.6.0). All SNVs and SVs detected in the designated off-target sites, estimated for individual gene-edited target sites, were counted and summarized (Supplementary Data 10).

## FISH assays

Samples were prepared following the protocols provided by Empire Genomics for FISH slide preparation and hybridization using SwiftFISH rapid hybridization buffer. The assay employed probes RP11-768C3, labeled with 5-Fluorescein dUTP, which specifically hybridizes to chr17:78091209-78262882 (GRCh37/hg19), and a chromosome 17 Control Probe labeled with 5-TAMRA dUTP, which

specifically hybridizes to the centromeric region of chromosome 17 (Empire Genomics). Fluorescent images were acquired using a DMI6000B microscope (Leica) using the Leica Application Suite Advanced Fluorescence software program (version 3.1; Leica). Signals were enhanced by level adjustment using Photoshop (version 24.6; Adobe). See the Source Data file for uncropped and unprocessed images.

## Statistics and reproducibility

No statistical method was used to predetermine the sample size. No data were excluded from the analyses. The experiments were not randomized. The investigators were not blinded to the allocation during the experiments and outcome assessment. All colony formation assays and long-range PCR were repeated in at least three independent experiments. Sanger sequencing, depicted in Fig. 1f, was performed once. Sanger sequencing to confirm gene editing efficiency was performed in three independent experiments (as groups). sgRNA pool screening tests using a cell proliferation assay (Fig. 2a) were conducted twice. All other proliferation assays were repeated in at least three independent experiments. AmpNGS analyses depicted in Fig. 2b were performed once, with 16 similar samples analyzed. Other AmpNGS analyses were repeated in three independent experiments. Immunoblotting experiments were repeated twice or three times. The statistical tests performed have been specified in the relevant figure legends. The proportion data containing 0 was adjusted using the following formula:

$$AdX = (X + 10^{-10})/(1 + 2 \times 10^{-10}), \quad (4)$$

where X is the original proportion, and AdX is the adjusted proportion. The data depicted in Fig. 7a (The theoretical upper limit was 0.5, while the actual maximum value was 0.526) were adjusted using the following formula:

$$AdY = Y/0.53, \quad (5)$$

where Y is the original proportion, and AdY is the adjusted proportion. The original proportion data and adjusted proportion data were transformed into a continuous scale using probit transformation in the Excel software program (version 2306; Microsoft). Two-tailed unpaired t-tests, a one-way analysis of variance (ANOVA) followed by Dunnett's or Tukey's multiple comparisons test (two-sided), or a two-way ANOVA with Šídák's or Tukey's multiple comparisons test (two-sided) were employed for statistical analyses as described in each figure legend. In Supplementary Fig. 3d, the Pearson correlation coefficient was computed (two-sided).

All statistical analyses were performed using the Prism software program (version 9.4; GraphPad). Graphs were created using the Excel (version 2306; Microsoft) or Prism (version 9.4; GraphPad) software program. The data represent the mean ± standard deviation (SD) from three independent experiments unless indicated otherwise in each figure legend.

## Reporting summary

Further information on research design is available in the Nature Portfolio Reporting Summary linked to this article.

## Data availability

The AmpNGS data generated in this study have been deposited in the NCBI BioSample database under accession codes SAMN36871804-SAMN36872143 and can be found under bioproject PRJNA1003052. The NGS data generated in this study have been deposited in the NCBI BioSample database under accession codes SAMN37027744-SAMN37027802 and can be found under bioproject PRJNA1006361. The other data generated in this study are provided in the Figures,

Supplementary Figures, and Source Data file. Source data are provided in this paper.

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

## Acknowledgements

We appreciate the help of Drs. M. Honma, M. Yasui, M. Kanemaki, K. Hirota, and F. Zhang with the materials. We thank Ms. H. Yasukawa and CentMeRE for their assistance with this study. This work was supported by the Japan Agency for Medical Research and Development (AMED) (JP17-19ek010929 [S.N.] and JP19-23am0401016 [S.N., H.S., and T.Og.]), Grants in Aid for Scientific Research KAKENHI (16K12596 [S.N.], JP19H04269 [S.N.], JP21K18314 [S.N.], and 22H03741 [S.N.]) from the Japan Society for the Promotion of Science, a grant from the Uehara Memorial Foundation [S.N.], Takeda Science Foundation [S.N.], Ichiro Kanehara Foundation [S.N.], The Mother and Child Health Foundation [S.N.] and Princess Takamatsu Cancer Research Fund [S.N.].

## Author contributions

S.N. designed the study. A.T. and S.N. conducted the majority of the experiments. H.S., S.N., and T.Ow. established the cell lines. T.Ow. performed long-range PCR, as shown in Fig. 4i. T.F. and T.Og. performed WGS analyses. M.S., Y.N., A.T., S.N., and T.Og. performed the ChIP-droplet digital PCR assay. S.N. wrote the paper. All authors read and approved the paper before submission.

## Competing interests

S.N. and A.T. contributed to the patent applications as follows: (1) patent applicant: Osaka University, name of inventor(s): Shinichiro NAKADA and Akiko TOMITA, application number: PCT/JP2019/031117 (National Phase entries, Japan), 2020-556604 (Examination, Japan), 201980075266.3 (Examination, China), 19885454.9 (Examination, Europe), and 17/294165 (Examination, US). Specific aspects of the manuscript covered in patent applications: NICER gene editing, SN, and AT. (2) Patent applicant: Osaka University; name of inventor: Shinichiro NAKADA; application number: PCT /JP2022/046917 (Publication, Japan). Specific aspects of the paper covered in the patent application: enhancement of NICER gene editing by gene knockout or knockdown, S.N. T.F. is an employee at Genomedia Inc. The remaining authors declare no competing interests.
