## [Peer Review File · Nature Communications]

Reviewers' Comments:

Reviewer #1:

Remarks to the Author:

General comments

In: "Multiple nicks-induced interhomolog homologous recombination corrects heterozygous mutations without DNA double-strand breaks and exogenous DNA" Tomita and colleagues study interhomolog recombination (IHR) in human cell lines triggered by tandem nicking using allele-specific and allele non-specific Cas9.D10A:gRNA complexes. This work involves the investigation of Cas9.D10A nickase and multiple guide RNAs (gRNAs), named primary (i.e., SNV/allele-specific) and secondary (i.e., SNV/allele non-specific) gRNAs as IHR-stimulating agents. The results expand on earlier and more recent findings from Davies et al (2020) (cited) and Roy et al., (2022) (not cited) showing that, respectively, DNA nicks induced by Cas9 nickases at both homologs induce IHR in somatic human cells and somatic Drosophila tissues. Specifically, by relying on the aforementioned multiple nicks (MNs), Tomita et al. report that IHR frequencies can be enhanced without substantial concomitant installation of unwanted genomic indels or gross rearrangements associated with Cas9 nuclease delivery.

The tracking of genome editing events at the clonal and population levels using independent assays is thorough. The resulting genotyping datasets together with data from experiments in cellular models with knocked-out or knocked-down DNA repair factors allowed the authors to put forward working models for nick-initiated IHR. Finally, their MN-IHR approach has potential for correcting genetic disorders caused by compound heterozygous mutations or, one can argue, autosomal dominant mutations provided that the efficiencies can be substantially improved and unwanted loss-of-heterozygosity byproducts are prevented.

References:

Davis et al. POLQ suppresses interhomolog recombination and loss of heterozygosity at targeted DNA breaks. Proc Natl Acad Sci USA. 2020 117:22900-22909. PMID: 32873648

Roy et al. Cas9/Nickase-induced allelic conversion by homologous chromosome-templated repair in Drosophila somatic cells. Sci Adv. 2022 PMID: 35776792

My major and minor comments together with specific questions are detailed next.

Major comments

Figure 1B. Under the column "TK1 exon 4 genotype", first/top entry, shouldn't the cyan allele be mutant instead of WT as indicated in Figure 1A for TK6261 cells? If so, please correct this and the other genotypes below accordingly.

Figure 1C. The diagram illustrating hybridization of the allele-specific gRNA spacer (red) to allele B, i.e., allele without 1 ins C (cyan) is highly misleading in that the 19 most 5'-terminal spacer nucleotides can hybridize with the corresponding complementary target sequence in allele B that has a canonical NGG PAM next to it. As presented, the alignment is shifted by 1 bp leading to "artificially" only 3 complementary nucleotides facing each other and forming 3 hydrogen bonds between spacer and protospacer sequences instead of the 19 possible.

The authors need to check the presumed allelic specificity by exposing cells to CRISPR complexes containing Cas9 nuclease (instead of Cas9.D10A nickase) and the relevant gRNA, i.e., sgTKex4_mt20s. If there is cleavage of allele B they need to reformulate their findings and, perhaps, conclusions.

Figure 3C. Similarly, the schematics depicting the hybridization of the allele-specific gRNA spacer (pink) to the, in this case, allele A, i.e., allele without 1-bp substitution (black) is also highly misleading in that, with the exception of 1 mismatch derived from the 1-bp substitution, 19 spacer nucleotides can hybridize with the corresponding complementary target sequence in allele A. However only 2 instead of 19 hydrogen bonds between spacer and protospacer sequences are shown heightening the presumed gRNA allelic specificity.

Indeed, as protospacers with single mismatches to spacer sequences can be readily susceptible to

cleavage (as initially shown, for instance, by Fu et al. High-frequency off-target mutagenesis induced by CRISPR-Cas nucleases in human cells. *Nat Biotechnol.* 2013 31:822-6. PMID: 23792628), it is possible that the allele-specific gRNA used in these experiments is not specific for allele B.

The authors need to check this by exposing cells to CRISPR complexes containing Cas9 nuclease (instead of Cas9.D10A nickase) and the relevant gRNA, i.e., sgEX5_mt28as/TR2. If there is cleavage of allele A they need to reformulate their findings and, perhaps, conclusions.

The near-diploid TK6 human lymphoblast cell line, and its clonal derivatives, were used in most experiments. For clarity (especially to non-experts) and in view of the importance of ploidy number in IHR studies, indicate the actual karyotype of this line so that, at a minimum, chromosome 17 number (TK1 target gene location), is provided.

Methods. The nomenclature of the TK6 cells with BRCA2 alleles (chr. 13) targeted with the auxin-regulated degron indicates biallelic targeting, however, further down it is stated that TK6 cells are trisomic for chr. 13. Is thus one of the alleles not targeted?

What is the ploidy of the other alleles targeted with the auxin-regulated degron?

Methods. Quantification and statistical analysis. Regarding the sentence: "Data represent the means \pm standard deviations of three experiments unless indicated otherwise in the figure legends."

If that is indeed the case indicate: "...three independent experiments..." or, alternatively, "... three biological replicates...".

Minor comments

The sentence: "Considering that the MN-IHR was below 0.2% when DSBs were generated on both homologs by Cas9 21, the MNs repair rate via IHR was markedly higher than expected." Is confusing in that MN-IHR does not involve DSB catalyses. Remove MN from the composite acronym MN-IHR?

Page 7. 3rd lane from end. I believe that Figure 3C should indicate Figure 3B instead. Please double-check.

Page 9. In "...whereas the three types of MNs did not generate SNVs..." For clarity, specify the 3 types of MNs.

Page 10. One can argue that owing to the intrinsically preliminary/speculative character of working models, the entire 1st paragraph of page 10 can best be placed in the Discussion section (possibly shortened or rephrased).

The same applies to the text in the 2nd paragraph of page 13 and, in this paragraph, to be consistent with the nomenclature in Supplementary Figure 14 substitute (1), (2), etc ... by (i), (ii), etc ...

In several places in the text the authors indicated the wavy line (approximately sign) preceding a value (e.g. %). In these cases, indicate instead the actual mean \pm SD or SEM (n=X) values.

Introduction, 1st paragraph. The sentence: "For example, a *Streptococcus pyogenes* Cas9 (spCas9)-guide RNA (gRNA) complex detects a target sequence adjacent to the protospacer adjacent motif sequence (PAM) "NGG," and..." is not correct in that the complex detects/interrogates firstly PAM sequences via the PIC domain of Cas9 and only after that the protospacer. Rephrase.

Introduction, 2nd paragraph. Perhaps "stability" is more appropriate in this context than "homeostasis".

Introduction. In: "...involving the use of DNA nicks (DNA damage in which a phosphodiester bond does not exist between the adjacent DNA molecules in one of the double strands of DNA)..."

The "does not exist" part is somewhat unorthodox phrasing and, in this context, "DNA molecules" should best read "nucleotides" for the sake of accuracy.

The important acronym HJ is not defined as Holliday junction.

At the end of page 11, for non-experts, please motivate the rationale for interfering with CtIP, BRCA1 and BRCA2 expression by briefly indicated their roles in DNA repair. If needed, dispense such information to the other factors as well.

The very short Results sub-section: "Repeated MNs induction increases the gene correction rate" should profit from a brief introductory description of the experimental set-up used so that the reader can understand the results presented without having to directly inspect to the respective Figures.

At first occasion, the authors should clearly state what discriminates the composite acronym MN-IHR from MN-IH-HR.

1st paragraph page 14. "Multiple nicking" (twice) can be abbreviated as MN.

Page 14. 3rd paragraph. Call to Figure 6D should be to Figure 6E. Double-check.

Page 14. The authors state that: "The FA18JTO hTERT cell line is derived from a patient with Fanconi anemia and has been transformed with the stable expression of hTERT" However, it is of notice that stable expression of hTERT normally results in immortalization not transformation (full-fledged or otherwise) of cells.

Page 14. The authors write: "We performed gene editing using two different experimental conditions (see: Methods section)." However, the phrasing "(see: Methods section)" is unorthodox and should thus be avoided, in my opinion. Specify instead for clarity.

Discussion. The sentence: "The therapeutic utility of CRISPR gene editing in embryos is a highly debated topic." Warrants one or more references.

Discussion. The linking of the sentence: "Some studies have indicated that frequent IH-HR enables administering gene therapy to patients with genetic diseases 7,42." to refs 7 and 42 is somewhat ambiguous. Rephrase for the sake of formal accuracy, e.g., "...might enable germ line gene therapies."

In the text gRNA names have the sg prefix whereas in the Figures the names often lack this prefix. Select a nomenclature and use it consistently throughout the manuscript.

Discussion. The sentence: "However, other studies have suggested that CRISPR gene editing utilizing DSBs causes chromosomal disruption by inducing large deletions or rearrangements 43-45"

On the basis of the data presented in the referred studies, I believe that "suggested" can be substituted by "demonstrated".

Methods. Provide the sequence of the tracrRNA (and crRNAs) used.

It is not clear to me what is the role of the cells with genotype PARP1-/-/PARP2-/+ in the study?

Methods. It is stated: "Supplementary Table 6 details the sgRNAs and crRNAs used in this study." However, it seems that the list only contains target sequences of sgRNAs.

Methods. In: "To assess colony formation, we followed a procedure described previously by 17." Name the authors of ref 17.

Reviewer #2:

Remarks to the Author:

It is getting a common view in the CRISPR community that DSB-dependent genome editing frequently induces undesired outcomes at on-target site such as large deletion and chromosomal rearrangement, and integration of vector sequences as well as p53 activation, thus the field is moving to DSB-independent approaches such as base and prime editing, Casposon, RNA editing, and integrase, especially for therapeutic purpose. In this manuscript, Tomita et al adds a new approach named MN-IH-HR, which utilizes multiple nicks and a homologous chromosome as a repair template to convert heterozygous mutation into wildtype, without DSB and exogenous repair template. The authors demonstrate 1) optimization of MN-IH-HR reaching significantly higher efficiency than previous described single nick-induced reciprocal exchange of homologs, 2) mechanistic insights of MN-IH-HR, 3) target specificity, 4) molecular basis, and 5) correction of human disease mutations. This is very interesting manuscript in terms of both DNA repair and biotechnology. Although there is a long-standing debate about whether IHR-mediated gene conversion occurs in mammalian zygotes, the authors utilized the idea of "self-repair" to fix somatic mutations and demonstrate distinct molecular basis of DSB-dependent and -independent IHR. I also find a potentially interesting application of IHR. This manuscript will be of broad interest to the readership of Nature Communications. Before publication, I suggest several points to further strengthen the author's claims and improve the manuscript.

Major comments

1. It is not clear to me how multiple nicks, especially on a same allele, promote IHR. The authors propose a model (Supplementary Fig. 14), however, the second nick on a same strand promotes MMR (step (v)), eliminating the newly formed wildtype strand (step (vi)). Thus, MN seems not contributing to these steps. [If the second nick is made on an opposite strand, the newly formed wildtype strand will be chosen to fix the mismatch (same as base and prime editing), thereby promoting IHR. This can be addressed by using H840A nickase for the second nick.] Then, how multiple nicks contribute to steps (ii) and (iii)? This is more complicated in the cases of three sgRNAs (Fig. 2D). Does MN-IH-HR require DNA replication/cell cycle progression? The authors should provide more insights. The author's discussion based on DSB (4th paragraph in Discussion) is not informative to explain MN-enhanced IHR.
2. Similarly, how donor and recipient alleles are determined? In the case of one primary nick on one allele and two secondary nicks on both alleles, the allele with two nicks may be a recipient allele. How about the cases of single nicks on both alleles (mut-4s+WT121S in Fig. 3A, Ex5_mt28as/TR2+E7_WT136as in Fig. 3D, and Ex4_mt20s+Ex5_mt-4s in Supplementary Fig. 3B)? Theoretically, Ex7 WT, Ex7 insG, and Ex5 del8 alleles can also be converted into insG, WT, and WT, respectively. Are these two sgRNA target sites (for example, del8 and insG in Fig3A) converted into homozygous together, or in a mutually exclusive manner? Are conversion directions same or opposite (for example, del8 becomes WT in Ex5 and insG becomes in WT in Ex7, or del8 becomes WT in Ex5 while WT becomes insG)? How about the cases of three sgRNAs (Fig. 2D)? The authors should perform SCC without CHAT selection followed by long PCR and cloning to determine genotypes of two sgRNA target sites on each allele.
3. The authors investigated conversion tract by testing various secondary sgRNAs and genotyping an SNV (Fig. 5). However, this is not enough to show the conversion tract length. The authors should determine the conversion tract length by genotyping multiple SNVs of SCC by using long PCR. Since the authors already have WGS data, it's easy to pick up SNV markers. The claim that the secondary nick on the donor allele limits the conversion tract length is very interesting. Unfortunately, the data is based on cell population and CHAT selection which biases the genotype. The authors should perform SCC without CHAT selection followed by long PCR and cloning to determine genotypes of multiple SNV along conversion tract.
4. The utility of MN-IH-HR to correct disease mutations is not convincing. Although the authors increased MN-IH-HR efficiency up to 18.5% at TK1 locus, it's significantly lower than the efficiencies of base editing or prime editing. Further, MN-IH-HR efficiencies seem very low in other three loci. It's only 2.7% at FANCA and even not shown at ADH5 and ATM. Immunoblots indicate very low levels of corrected gene expressions and it's not shown whether these low levels of gene correction are enough to ameliorate disease symptoms. Thus, the mutations tested in here such as

SNV and short deletions are not the good therapeutic targets for MN-IH-HR compared to base editing and prime editing. Instead, considering the long conversion tract of MN-IH-HR, large copy number variants (CNVs) and chromosomal rearrangement such as chromothripsis may be good targets. Base and prime editing cannot correct these variations (to be fair, prime editing potentially can, by combining recombinase with exogenous donor). If the authors show an example to correct CNV or chromosomal rearrangement, it would be great.

Minor comments

1. Addition of distances between nicks, mutations, and exons on all the schematics are helpful.
2. Schematic for Table 3 is helpful.
3. Allele specificities of allele-specific sgRNAs are not fully shown.
4. The last paragraph regarding stem cell in the Discussion section is not relevant to this work at all.

Reviewer #3:

Remarks to the Author:

The authors present a new gene editing approach that entails the introduction of DNA nicks at the mutated gene allele to trigger gene correction (conversion) directed by a homologous chromosome. This system is suggested to have tangible benefits relative to traditional CRISPR-based gene editing techniques, as mutation correction is characterized by a reduced propensity of secondary mutations as DNA double-strand breaks are, apparently, not induced. The authors demonstrate gene conversion by measuring the recovery of TK1 activity upon introducing one or more targeted DNA nicks. They also find that nick-induced gene conversion has a lesser dependence on canonical homologous recombination factors, such as BRCA1 and BRCA2. Finally, the system was used to correct mutations in known hereditary disease-derived cells. While nick-induced gene conversion seems like a promising step forward in gene editing, much of what is proposed regarding mechanism is inferred from the data without direct evidence. This is a significant flaw. There is also a possibility that the nicks are converted to one-ended DSBs that are repaired via HR.

Specific Comments

1. The conclusions drawn rely on the premise that multiple nicks are induced and that they are targeted to the correct loci. Can the authors demonstrate this is the case, specifically when they describe targeting one, but not both alleles?
2. Conclusions pertaining to the mechanisms and pathways involved need further support. For example, how are the long stretches of ssDNA generated after multiple nicks are presented? EXO1 is involved in both MMR and HR. Does knockdown of EXO1 reduce the MN-IH-HR efficiency (via ssDNA generation), or are there other nucleolytic machineries (e.g. BLM-DNA2) involved in ssDNA generation?
3. Also, if ssDNA stretches are generated, RPA foci should be observable, and they may correlate to the distance between the nicks. Is this true?
4. Although the authors do not knockout or inhibit RAD51, they need to compare RAD51 foci after DSB- or nick-induced gene editing.
5. Supplementary Figure 13 A-C show that CtIP-AID, and BRCA1-AID cells have significantly less CtIP and BRCA1 expression compared to TSCER2 cells, even without auxin treatment, indicating these are leaky degrons.
6. In the Discussion, the author state that "MNs are repaired in absence of BRCA2", however, the repair was significantly reduced with BRCA2 depletion as shown in 6D. Please clarify.
7. Depletion of XRCC1 and PARP1 increases MN-IH-HR suggesting SSB repair pathway completes with this mechanism, as shown in Supplemental Fig 13K. Since the authors have not presented any data demonstrating that MNs are really generated at the said loci, it would be helpful to confirm the same via XRCC1 and PARP1 ChIP studies.
8. The nicks could be converted to one-ended DSB in dividing cells that would be repaired via HR. Repeated introduction on nicks increases the gene conversion event, which could be via increasing the possibility of generation of one-ended DSB, instead of repair via SSB repair pathway. Furthermore, reduction of gene conversion events when the nicks were introduced in the other allele, as shown in Figure 5, may also indicate this to be true. Discuss this point at the minimum.

Minor Comments

1. The title is cumbersome and not a simple representation of the work. Suggestion: "Inducing multiple nicks promotes interhomolog recombination to correct heterozygous mutations without DNA double-strand breaks."
2. Labelling of Supplementary Figure 13 F and G are swapped.

Point-by-point response

Tomita et al. Inducing multiple nicks promotes interhomolog homologous recombination to correct heterozygous mutations in somatic cells

We would like to express our gratitude to the editors and reviewers for the thorough review of our manuscript and the constructive comments provided, which have greatly contributed to the improvement of our work. In accordance with the reviewer's comments, we have conducted numerous additional experiments and carefully revised the manuscript, incorporating the reviewer's suggestions into the revised version. As a result, we believe that the paper has been significantly improved. Below, we provide point-by-point responses to each of the comments from the reviewers. The text in green is the original comment from the reviewer, while the text in black or red is the author's response. We appreciate the opportunity to have our manuscript reconsidered for publication in Nature Communications.

Reviewer #1 (Remarks to the Author):

General comments

In: "Multiple nicks-induced interhomolog homologous recombination corrects heterozygous mutations without DNA double-strand breaks and exogenous DNA" Tomita and colleagues study interhomolog recombination (IHR) in human cell lines triggered by tandem nicking using allele-specific and allele non-specific Cas9.D10A:gRNA complexes. This work involves the investigation of Cas9.D10A nickase and multiple guide RNAs (gRNAs), named primary (i.e., SNV/allele-specific) and secondary (i.e., SNV/allele non-specific) gRNAs as IHR-stimulating agents. The results expand on earlier and more recent findings from Davies et al (2020) (cited) and Roy et al., (2022) (not cited) showing that, respectively, DNA nicks induced by Cas9 nickases at both homologs induce IHR in somatic human cells and somatic Drosophila tissues. Specifically, by relying on the aforementioned multiple nicks (MNs), Tomita et al. report that IHR frequencies can be enhanced without substantial concomitant installation of unwanted genomic indels or gross rearrangements associated with Cas9 nuclease delivery.

The tracking of genome editing events at the clonal and population levels using independent assays is thorough. The resulting genotyping datasets together with data from experiments in cellular models with knocked-out or knocked-down DNA repair factors allowed the authors to put forward working models for nick-initiated IHR. Finally, their MN-IHR approach has potential for correcting genetic disorders caused by compound heterozygous mutations or, one can argue, autosomal dominant mutations provided that the efficiencies can be substantially improved and unwanted loss-of-heterozygosity byproducts are prevented.

References:

Davis et al. POLQ suppresses interhomolog recombination and loss of heterozygosity at targeted DNA breaks. Proc Natl Acad Sci USA. 2020 117:22900-22909. PMID: 32873648

Roy et al. Cas9/Nickase-induced allelic conversion by homologous chromosome-templated repair in Drosophila somatic cells. Sci Adv. 2022 PMID: 35776792

My major and minor comments together with specific questions are detailed next.

We would like to express our gratitude for the positive evaluation of our research findings. We also appreciate the valuable suggestions provided, including proposals for an important paper to be cited (Roy et al., Sci Adv. 2022, which is now cited in the new manuscript as ref 24), improvements to figures, and a wide range of constructive feedback. We have addressed all of the comments made by Reviewer #1 in our response.

Major comments

Figure 1B. Under the column “TK1 exon 4 genotype”, first/top entry, shouldn't the cyan allele be mutant instead of WT as indicated in Figure 1A for TK6261 cells? If so, please correct this and the other genotypes below accordingly.

Thank you for your comments. We have revised the original figure, as it was prone to causing misunderstandings. To prevent confusion among readers, we have provided more detailed information in the new **Figure 1C**.

Figure 1C. The diagram illustrating hybridization of the allele-specific gRNA spacer (red) to allele B, i.e., allele without 1 ins C (cyan) is highly misleading in that the 19 most 5'-terminal spacer nucleotides can hybridize with the corresponding complementary target sequence in allele B that has a canonical NGG PAM next to it. As presented, the alignment is shifted by 1 bp leading to “artificially” only 3 complementary nucleotides facing each other and forming 3 hydrogen bonds between spacer and protospacer sequences instead of the 19 possible.

The authors need to check the presumed allelic specificity by exposing cells to CRISPR complexes containing Cas9 nuclease (instead of Cas9.D10A nickase) and the relevant gRNA, i.e., sgTKex4_mt20s. If there is cleavage of allele B they need to reformulate their findings and, perhaps, conclusions.

Thank you for your comments. We would like to address your concerns in the following response.

We electroporated sgTKex4_mt20s and wild-type Cas9 mRNA into TK6261 cells and extracted genomic DNA for analysis. Amplicon-based NGS analysis was performed on the region surrounding *TK1* exon 4. Our results showed that $48.5 \pm 4.1\%$ of all reads corresponded to the wild-type exon 4 sequence (allele B), while the proportion of reads containing c.231_232insC but no other mutations (originating from allele A) was $2.7 \pm 2.0\%$ (New Supplementary Figure 1E). These findings suggest that sgTKex4_mt20s specifically targets the mutant sequence on allele A. We apologize for not including this data explicitly in the manuscript before submission.

Regarding the figure, we have revised it by adding dashed lines to represent the base pair formations that occur when the single base insertion is ignored. Additionally, we have included the base pair formations that occur when CGG is used as the PAM, indicated by solid green lines (New Figure 1D).

Thank you again for your helpful comments.

Figure 3C. Similarly, the schematics depicting the hybridization of the allele-specific gRNA spacer (pink) to the, in this case, allele A, i.e., allele without 1-bp substitution (black) is also highly misleading in that, with the exception of 1 mismatch derived from the 1-bp substitution, 19 spacer nucleotides can hybridize with the corresponding complementary target sequence in allele A. However only 2 instead of 19 hydrogen bonds between spacer and protospacer sequences are shown heightening the presumed gRNA allelic specificity.

Indeed, as protospacers with single mismatches to spacer sequences can be readily susceptible to cleavage (as initially shown, for instance, by Fu et al. High-frequency off-target mutagenesis induced by CRISPR-Cas nucleases in human cells. Nat Biotechnol. 2013 31:822-6. PMID: 23792628), it is possible that the allele-specific gRNA used in these experiments is not specific for allele B.

The authors need to check this by exposing cells to CRISPR complexes containing Cas9 nuclease (instead of Cas9.D10A nickase) and the relevant gRNA, i.e., sgEX5_mt28as/TR2. If there is cleavage of allele A they need to reformulate their findings and, perhaps, conclusions.

We conducted experiments to confirm the specificity of sgRNAs, as described below.

- sgEx5_mt28as/TR2 (Result: allele-specific)

We introduced sgEx5_mt28as/TR2 and wild-type Cas9 into TSCER2 cells and performed amplicon-based NGS analysis on the region surrounding exon 5. Our results showed that among all reads, $58.7 \pm 4.8\%$ were the wild-type exon 5 sequence (originating from allele A), while the proportion of reads containing c.326G>A but no other mutations (originating from allele B) was $0.2 \pm 0.3\%$ (New Supplementary Figure 3G). These findings indicate that sgEx5_mt28as/TR2 specifically targets the mutant sequence on allele B.

- sgEx7_WT136as (Result: not allele-specific)

We introduced sgEx7_WT136as and wild-type Cas9 into TSCER2 cells and performed amplicon-based NGS analysis on the region surrounding exon 7. However, we did not observe allele specificity in this experiment using sgEx7_WT136.

Therefore, we have removed the experimental data using sgEx7_WT136as or sgEx7_mt136as from the previous Figure 3D. We are using the remaining data to show only that MN-IH-HR can be induced by MNs on the noncoding strand (New Figure 3A-C).

To address the gene correction efficiency when each allele is nicked once, we created TK6261_S43v11 and TK6261_S43v44 cells and performed the experiments. The results are shown in the new Figure 3D, E.

- sgEx5_mt-4s (Result: allele-specific)

We introduced sgEx5_mt-4s and wild-type Cas9 into TK6261 cells and performed amplicon-based NGS analysis on the region surrounding exon 5. Our results, shown in New Supplementary Figure 3C, demonstrate allele specificity for sgEx5_mt-4s.

- sgEx7_mt121s (Result: allele-specific)
- sgEx7_WT121s (Result: not allele-specific)

We introduced either sgEx7_mt121s or sgEx7_WT121s and wild-type Cas9 into TK6261 cells and performed amplicon-based NGS analysis on the region surrounding exon 7. We found that sgEx7_mt121s was an allele-specific sgRNA, while sgEx7_WT121s did not show allele specificity.

We have removed these data from the revised manuscript. We are using the remaining data to demonstrate that MNs can repair the 8bp deletion (New Supplementary Figure 3A, B).

The near-diploid TK6 human lymphoblast cell line, and its clonal derivatives, were used in most experiments. For clarity (especially to non-experts) and in view of the importance of ploidy number in IHR studies, indicate the actual karyotype of this line so that, at a minimum, chromosome 17 number (TK1 target gene location), is provided.

We have included the karyotype of TK6 cells (47, XY, +13) in the final paragraph of Page 4. Additionally, we have clarified that the TK6261 cells contain two copies of the long arm of chromosome 17 using the two-color FISH method (New Figure 1B).

Methods. The nomenclature of the TK6 cells with BRCA2 alleles (chr. 13) targeted with the auxin-regulated degron indicates biallelic targeting, however, further down it is stated that TK6 cells are trisomic for chr. 13. Is thus one of the alleles not targeted?

What is the ploidy of the other alleles targeted with the auxin-regulated degron?

We apologize for the confusion regarding our notation of BRCA2^{BRCA2-AID/BRCA2-AID}. As described in the **Methods section**, BRCA2^{BRCA2-AID/BRCA2-AID/BRCA2-AID} cells are generated by replacing three copies of the BRCA2 gene with BRCA2-AID (+ selection marker) using three selection markers (*NEOMYCIN*^R, *HYGROMYCIN*^R, and *HISTIDINO*^R). We have corrected the notation in the manuscript (**the final paragraph, of Page 19**) and figures (**Figure 6B, Supplementary Figure 11B**) to BRCA2^{BRCA2-AID/BRCA2-AID/BRCA2-AID}. BCRA1 (Chr. 17) and CtIP (Chr. 18) are biallelic.

Methods. Quantification and statistical analysis. Regarding the sentence: "Data represent the means ± standard deviations of three experiments unless indicated otherwise in the figure legends." If that is indeed the case indicate: "...three independent experiments..." or, alternatively, "... three biological replicates..."

Thank you for your comment. We have revised the text as follows: " Data represent the means ± standard deviations of three independent experiments unless indicated otherwise in the Figure legends." (**Page 27**)

Minor comments

The sentence: "Considering that the MN-IHR was below 0.2% when DSBs were generated on both homologs by Cas9 21, the MNs repair rate via IHR was markedly higher than expected." Is confusing in that MN-IHR does not involve DSB catalyses. Remove MN from the composite acronym MN-IHR?

We appreciate your comment; however, the sentence in question has been removed from the revised manuscript.

Page 7. 3rd lane from end. I believe that Figure 3C should indicate Figure 3B instead. Please double-check.

In the revised manuscript, we have labeled the corresponding figure as **Figure 3A**.

Page 9. In “...whereas the three types of MNs did not generate SNVs...” For clarity, specify the 3 types of MNs.

We have made the following changes to the revised manuscript on Page 13: "... while the combination of Nickase, sgEx4_mt20s, and one of sgS14, sgS8, or sgS34 did not generate SNVs or small indels..." (Page13)

Page 10. One can argue that owing to the intrinsically preliminary/speculative character of working models, the entire 1st paragraph of page 10 can best be placed in the Discussion section (possibly shortened or rephrased).

The same applies to the text in the 2nd paragraph of page 13 and, in this paragraph, to be consistent with the nomenclature in Supplementary Figure 14 substitute (1), (2), etc ... by (i), (ii), etc ...

Thank you for your feedback. We have moved the description of the working models to the Discussion section.

Due to limited space, detailed information regarding **Supplementary Figure 15B** (previously Supplementary Figure 14) has been omitted from the Results and Discussion sections of the revised manuscript. However, a detailed description of this figure can be found in the figure legend.

In several places in the text the authors indicated the wavy line (approximately sign) preceding a value (e.g. %). In these cases, indicate instead the actual mean +/- SD or SEM (n=X) values.

We have made the changes as Reviewer #1 suggested, otherwise the actual mean +/- SD or SEM (n=X) values are indicated in figures.

*Introduction, 1st paragraph. The sentence: “For example, a *Streptococcus pyogenes* Cas9 (spCas9)–guide RNA (gRNA) complex detects a target sequence adjacent to the protospacer adjacent motif sequence (PAM) “NGG,” and...” is not correct in that the complex detects/interrogates firstly PAM sequences via the PIC domain of Cas9 and only after that the protospacer. Rephrase.*

Thank you for pointing out the error. Due to the word limit of the manuscript, we were unable

to provide a detailed description. We have revised the text as follows: "A *Streptococcus pyogenes* Cas9 (Cas9)–guide RNA (gRNA) complex cleaves both DNA strands via its two nuclease domains to generate a two-ended DNA double-strand break (DSB) in the protospacer sequence 5' to the protospacer adjacent motif (PAM) sequence “NGG”^{1, 2, 3.}” (The first paragraph, Page 3)

Introduction, 2nd paragraph. Perhaps “stability” is more appropriate in this context than “homeostasis”.

We have made the changes as you suggested. (Second paragraph, Page 3)

Introduction. In: “...involving the use of DNA nicks (DNA damage in which a phosphodiester bond does not exist between the adjacent DNA molecules in one of the double strands of DNA)...”
The “does not exist” part is somewhat unorthodox phrasing and, in this context, “DNA molecules” should best read “nucleotides” for the sake of accuracy.

In the revised manuscript, we have written, " Nicks are a type of DNA damage where a phosphodiester bond between adjacent nucleotides is absent in one strand of the DNA double-helix,..." (The final paragraph, Page 3)

The important acronym HJ is not defined as Holliday junction.

We defined HJ as Holliday junctions (The final paragraph, Page 11).

At the end of page 11, for non-experts, please motivate the rationale for interfering with CtIP, BRCA1 and BRCA2 expression by briefly indicated their roles in DNA repair. If needed, dispense such information to the other factors as well.

We described the roles of DNA repair proteins, such as CtIP, BRCA1, BRCA2, etc., on Pages 11-13.

The very short Results sub-section: “Repeated MNs induction increases the gene correction rate” should profit from a brief introductory description of the experimental set-up used so that the reader can understand the results presented without having to directly inspect to the respective Figures.

We provided details of the experiments on Pages 13-14.

At first occasion, the authors should clearly state what discriminates the composite acronym MN-IHR from MN-IH-HR.

In the revised manuscript, we have written the following: "There are two types of IHR: interhomolog end-joining (IH-EJ) and interhomolog homologous recombination (IH-HR). IH-EJ joins DNA ends on homologous chromosomes, leading to an exchange of chromosome arms that can potentially alleviate the disease phenotype by converting alleles with compound heterozygous mutations to a wild-type allele and an allele with two mutations. IH-HR repairs DNA breaks via HR, using a homologous chromosome as a repair template, and can correct single heterozygous mutations or one of the compound heterozygous mutations to the wild-type sequence." (The first paragraph, Page 4)

1st paragraph page 14. “Multiple nicking” (twice) can be abbreviated as MN.

We have abbreviated "Multiple nicking" as "MNing." (Final paragraph, Page 5)

Page 14. 3rd paragraph. Call to Figure 6D should be to Figure 6E. Double-check.

In the revised manuscript, this is now Supplementary Figure 12A.

Page 14. The authors state that: “The FA18JTO hTERT cell line is derived from a patient with Fanconi anemia and has been transformed with the stable expression of hTERT” However, it is of notice that stable expression of hTERT normally results in immortalization not transformation (full-fledged or otherwise) of cells.

We have revised the sentence to read, "The FA18JTO hTERT cell line is derived from a patient with Fanconi anemia and has been immortalized with the stable expression of hTERT." (Third paragraph, Page 14)

Page 14. The authors write: "We performed gene editing using two different experimental conditions (see: Methods section)." However, the phrasing "(see: Methods section)" is unorthodox and should thus be avoided, in my opinion. Specify instead for clarity.

We have revised the sentence to read, "We performed gene editing using two different experimental conditions (total 3.6 pmol/μL of three sgRNAs + 179 ng/μL of Nickase mRNA or total 5.0 pmol/μL of three sgRNAs + 136 ng/μL of Nickase mRNA)." (Third paragraph, Page 14)

Discussion. The sentence: "The therapeutic utility of CRISPR gene editing in embryos is a highly debated topic." Warrants one or more references.

We cited the reference "Ledford H. 'CRISPR babies' are still too risky, says influential panel. *Nature*, doi.org/10.1038/d41586-020-02538-4 (2020)" (Third paragraph, Page 15)

Discussion. The linking of the sentence: "Some studies have indicated that frequent IH-HR enables administering gene therapy to patients with genetic diseases 7,42." to refs 7 and 42 is somewhat ambiguous. Rephrase for the sake of formal accuracy, e.g., "...might enable germ line gene therapies."

We revised the sentence as "Some have proposed that frequent IH-HR might enable germ line gene therapies^{46,47}," (Third paragraph, Page 15)

In the text gRNA names have the sg prefix whereas in the Figures the names often lack this prefix. Select a nomenclature and use it consistently throughout the manuscript.

Now, all the sgRNA names have the prefix throughout the manuscript.

Discussion. The sentence: "However, other studies have suggested that CRISPR gene editing utilizing DSBs causes chromosomal disruption by inducing large deletions or rearrangements 43-45"

On the basis of the data presented in the referred studies, I believe that “suggested” can be substituted by “demonstrated”.

We revised the sentence as “while others have demonstrated that CRISPR-mediated gene editing via DSBs can cause chromosomal disruption, such as large deletions or rearrangements^{48, 49, 50}.” (Third paragraph, Page 15)

Methods. Provide the sequence of the tracrRNA (and crRNAs) used.

The sequences of the tracrRNA and sgRNAs are provided in Method section and Supplementary table 1.

It is not clear to me what is the role of the cells with genotype PARP1-/-/PARP2-/+ in the study?

We have recognized that it is challenging to provide a logical explanation for the changes in MN-IH-HR efficiency caused by PARP1 knockout, given the diverse roles of PARP1 and PARP2 in DNA repair. Therefore, in the revised manuscript, we have chosen to exclude the data obtained from PARP1-/-/PARP2-/+ cells.

Methods. It is stated: “Supplementary Table 6 details the sgRNAs and crRNAs used in this study.” However, it seems that the list only contains target sequences of sgRNAs.

The sequences of the tracrRNA, crRNA and sgRNAs is provided in Method section and Supplementary table 1.

Methods. In: “To assess colony formation, we followed a procedure described previously by 17.” Name the authors of ref 17.

We revised the sentence as “previously described by Nakajima et al¹⁹.” (Final paragraph, Page 21)

Reviewer #2 (Remarks to the Author):

It is getting a common view in the CRISPR community that DSB-dependent genome editing frequently induces undesired outcomes at on-target site such as large deletion and chromosomal rearrangement, and integration of vector sequences as well as p53 activation, thus the field is moving to DSB-independent approaches such as base and prime editing, Casposon, RNA editing, and integrase, especially for therapeutic purpose. In this manuscript, Tomita et al adds a new approach named MN-IH-HR, which utilizes multiple nicks and a homologous chromosome as a repair template to convert heterozygous mutation into wildtype, without DSB and exogenous repair template. The authors demonstrate 1) optimization of MN-IH-HR reaching significantly higher efficiency than previous described single nick-induced reciprocal exchange of homologs, 2) mechanistic insights of MN-IH-HR, 3) target specificity, 4) molecular basis, and 5) correction of human disease mutations. This is very interesting manuscript in terms of both DNA repair and biotechnology. Although there is a long-standing debate about whether IHR-mediated gene conversion occurs in mammalian zygotes, the authors utilized the idea of “self-repair” to fix somatic mutations and demonstrate distinct molecular basis of DSB-dependent and -independent IHR. I also find a potentially interesting application of IHR. This manuscript will be of broad interest to the readership of Nature Communications. Before publication, I suggest several points to further strengthen the author’s claims and improve the manuscript.

Thank you very much for positively evaluating our research and for providing helpful suggestions to strengthen and further develop the arguments presented in this study. We appreciate your feedback and would like to respond to the comments made by Reviewer #2 as follows.

Major comments

1. It is not clear to me how multiple nicks, especially on a same allele, promote IHR. The authors propose a model (Supplementary Fig. 14), however, the second nick on a same strand promotes MMR (step (v)), eliminating the newly formed wildtype strand (step (vi)). Thus, MN seems not contributing to these steps. [If the second nick is made on an opposite strand, the newly formed wildtype strand will be chosen to fix the mismatch (same as base and prime editing), thereby promoting IHR. This can be addressed by using H840A nickase for the second nick.] Then, how multiple nicks contribute to steps (ii) and (iii)? This is more complicated in the cases of three sgRNAs (Fig. 2D). Does MN-IH-HR require DNA replication/cell cycle progression? The authors should provide more insights. The author’s discussion based on DSB (4th paragraph in Discussion) is not informative to explain MN-enhanced IHR.

Thank you for your valuable comments. Now we provide our proposed model in Supplementary Figure 15 and in Discussion sections. Briefly, we propose that at least two pathways are involved in MN-IH-HR: Replication-dependent (BRCA1- and BRCA2-dependent) pathway and replication-independent pathway (Supplementary Figure 15A). Multiple nicking expands exposed ssDNA region, to which RPA binds (Figure 5B). Therefore, multiple nicking promote IH-HR.

- Regarding “the old Supplementary Figure 13 (Supplementary Figure 15B in the revised manuscript)”:

In the DNA replication-independent pathway of MN-IH-HR, we expect the coding strand of the recipient allele (allele A) containing the nick to invade the donor allele (allele B) and anneal to the noncoding strand of allele B. Subsequently, DNA synthesis occurs on allele B before the coding strand of allele A re-anneals with the noncoding strand of allele A. In this case, the mutation on the coding strand is corrected, while the mutation on the noncoding strand is not, leading to the formation of a mismatch. The former Supplementary Figure 13 explains how the mismatches generated during the MN-IH-HR process are repaired (in this case, the corrected nucleotide is reverted to the mutant) based on the data from MSH2^{-/-}, MLH1^{-/-}, or MLH3^{-/-} cells shown in the new Figure 6E. The former Supplementary Figure 13 (now Supplementary Figure 15B) does not explain how multiple nicks (MNs) stimulate MN-IH-HR. In the revised manuscript, we aimed to avoid confusion and presented our proposed model for MN-IH-HR first (Supplementary Figure 15A), followed by the MMR model (Supplementary Figure 15B). Furthermore, we discussed the mechanism for MN-IH-HR in the Discussion section.

- Regarding “Cas9^{H840A}”:

It is impossible to generate only the secondary nick(s) using Cas9^{H840A} while creating the primary nick with Cas9^{D10A} in our experimental setting. If both primary and secondary nicks are generated using Cas9^{H840A}, the secondary nick would eventually occur on the repaired DNA strand (in this case, on the noncoding strand). Therefore, the data from experiments using Cas9^{H840A} provide no further scientific implications compared to that using Cas9^{D10A}.

Instead of using Cas9^{H840A}, we performed experiments below:

[Redacted]

We believe this result addresses the concerns of Reviewer 2. Since this data was obtained from another ongoing research project, we have decided not to include it in this paper.

• Regarding “replication and cell cycle progression”:

We employed palbociclib, a CDK4/CDK6 inhibitor, to arrest cells in the G1 phase before generating MNs. The efficiency of MN-IH-HR in cells treated with Palbociclib was approximately 45% compared to that in cells treated with DMSO, indicating that replication or cell cycle progression is not essential for MN-IH-HR (New Figure 5A, B, Supplementary Figure 10). Had replication or cell cycle progression been crucial for MN-IH-HR, we would have anticipated a significantly lower efficiency of MN-IH-HR in Palbociclib-treated cells. Our findings strongly suggest that MN-IH-HR operates throughout the cell cycle.

Additionally, our data presented in Figure 6A and B demonstrate that about 50% of asynchronously growing cells depend on BRCA1 and BRCA2 for MN-IH-HR. This observation cannot be accounted for by a unilateral mechanism, implying that both BRCA1 and BRCA2-dependent and independent pathways might contribute to MN-IH-HR (Supplementary Figure 15 A). One potential pathway reliant on BRCA1 and BRCA2 involves the repair of one-ended DSBs. In cycling cells, replication forks occasionally collide with a nick, potentially forming one-ended DSBs during replication. This pathway would be active in the S phase. Given that BRCA1 and BRCA2 are vital factors for HR, it is suggested that MN-IH-HR, independent of BRCA1 and BRCA2, does not involve nick-to-DSB conversion. Prior studies revealing the presence of BRCA2 and RAD51-independent nick-induced HDR (Davis et al., Cell Rep 2016) further support this notion. These results suggest that nicks generated by Nickase in the G1 phase are directly repaired by HR-like recombination.

We have thoroughly discussed in the Introduction and Discussion sections.

2. Similarly, how donor and recipient alleles are determined? In the case of one primary nick on one allele and two secondary nicks on both alleles, the allele with two nicks may be a recipient allele. How about the cases of single nicks on both alleles (mut-4s+WT121S in Fig. 3A, Ex5_mt28as/TR2+E7_WT136as in Fig. 3D, and Ex4_mt20s+Ex5_mt-4s in Supplementary Fig. 3B)? Theoretically, Ex7 WT, Ex7 insG, and Ex5 del8 alleles can also be converted into insG, WT, and WT, respectively. Are these two sgRNA target sites (for example, del8 and insG in Fig3A) converted into homozygous together, or in a mutually exclusive manner? Are conversion directions same or opposite (for example, del8 becomes WT in Ex5 and insG becomes in WT in Ex7, or del8 becomes WT in Ex5 while WT becomes insG)? How about the cases of three sgRNAs (Fig. 2D)? The authors should perform SCC without CHAT selection followed by long PCR and cloning to determine genotypes of two sgRNA target sites on each allele.

In the experiments using sgEx4_mt20s, sgS14 (or sgS8), and Nickase to selectively generate a nick only in TK1 exon 4 on the mutant allele (allele A), we observed cells with WT/WT exon 4, while no cells with c231_232insC/c231_232insC exon 4 were detected (Table 1; note that the SCCs were established in CHAT-free standard culture medium). These results demonstrate that, in a local region where a nick is present only in one allele, the allele containing the nick functions as the recipient, while the other serves as the donor. When employing sgEx4_mt21/WT20s (non-allele-specific), sgS14, and Nickase to produce nicks in TK1 exon 4 in both the mutant and healthy alleles, we found cells with WT/WT exon 4 and cells with c.231_232insC/c.231_232insC exon 4 (Table 1, second paragraph, Page 8). This observation suggests that when nicks are present in both alleles, either allele can act as both recipient and donor. In these experimental setups, the nick occurs only in the coding strand, leaving the noncoding strand intact. Consequently, there is no logical inconsistency in both alleles functioning as recipients and donors simultaneously.

Furthermore, by utilizing sgEx4_mt20s, sgEx5_mt-4s, and Nickase, we generated single nicks in TK1 exon 4 only on allele A and in TK1 exon 5 only on allele B. We observed cells with both exon 4 and exon 5 as WT homozygotes, as well as cells with either exon 4 or exon 5 as WT homozygote and the other as a heterozygote. However, cells with mutant homozygote exon 4 or mutant homozygote exon 5 did not appear (Table 1). This data implies that in the local region surrounding each nick, the nicked allele acts as the donor, and the allele without the nick serves as the recipient. As no further information was anticipated from the long-range PCR experiment followed by TA cloning (or Topo cloning), we did not perform the long-range PCR-TA cloning experiment (Please note that in TK6261 cells, no SNV other

than the SNV in exon 7 exists). Nevertheless, in other experimental settings, we conducted the long-range PCR-Topo cloning experiments (Figure 4C, D).

We have added the following statement to the Discussion section: In a local region where a nick is present only in one allele, the allele containing the nick functions as the recipient, while the other serves as the donor (Page 15).

3. The authors investigated conversion tract by testing various secondary sgRNAs and genotyping an SNV (Fig. 5). However, this is not enough to show the conversion tract length. The authors should determine the conversion tract length by genotyping multiple SNVs of SCC by using long PCR. Since the authors already have WGS data, it's easy to pick up SNV markers.

The claim that the secondary nick on the donor allele limits the conversion tract length is very interesting. Unfortunately, the data is based on cell population and CHAT selection which biases the genotype. The authors should perform SCC without CHAT selection followed by long PCR and cloning to determine genotypes of multiple SNV along conversion tract.

Unfortunately, no other SNV were identified on the *TK1* gene besides the neutral SNV in exon 7 described in the manuscript. Thus, we generated cells with an SNV in intron 5 of the *TK1* gene on allele B using a Base Editor and sgEx6_-92s (Figure 4C). As demonstrated in Figure 4C and 4D, we established 714 SCCs in CHAT-free medium after generating multiple nicks using sgEx5_mt-4s, sgS12, and Nickase. Among these, 11 SCCs were WT homo/hemi-zygotes for exon 5, while the remaining SCCs were heterozygous for exon 5. No cells with homo/hemi-zygous mutations (c.311_318del8) were detected.

We performed Sanger sequencing of *TK1* exon 4, intron 5, and exon 7 in 9 SCCs, where exon 5 was demonstrated to be WT homozygous. In 7 of these SCCs, exon 4 remained heterozygous, while intron 5 and exon 7 were WT homozygous (Figure 4D).

Next, in response to Reviewer 2's request, we conducted long-range PCR to include the target sequences of sgEx5_mt-4s and sgS12 in the PCR products. This was followed by TOPO-cloning and genotyping. The results revealed that the DNA sequence of allele B was replaced with that of allele A from exon 5 to exon 7 in 7 SCCs. Meanwhile, no SCCs with homo/hemi-zygous mutations (SNV) in intron 5 and/or exon 7 were found (Supplementary Figure 7). In summary, although not occurring in all cells, MNs can induce MN-IH-HR with long tract gene conversion.

4. The utility of MN-IH-HR to correct disease mutations is not convincing. Although the authors increased MN-IH-HR efficiency up to 18.5% at TK1 locus, it's significantly lower than the efficiencies of base editing or prime editing. Further, MN-IH-HR efficiencies seem very low in other three loci. It's only 2.7% at FANCA and even not shown at ADH5 and ATM. Immunoblots indicate very low levels of corrected gene expressions and it's not shown whether these low levels of gene correction are enough to ameliorate disease symptoms. Thus, the mutations tested in here such as SNV and short deletions are not the good therapeutic targets for MN-IH-HR compared to base editing and prime editing. Instead, considering the long conversion tract of MN-IH-HR, large copy number variants (CNVs) and chromosomal rearrangement such as chromothripsis may be good targets. Base and prime editing cannot correct these variations (to be fair, prime editing potentially can, by combining recombinase with exogenous donor). If the authors show an example to correct CNV or chromosomal rearrangement, it would be great.

We appreciate the valuable suggestions. Due to the unavailability of ideal model cell lines with CNVs or chromothripsis, we conducted gene editing in the TK6261_LD407E cells, which were generated in another ongoing research project. The TK6261_LD407E cells have a 662 bp deletion and 1 bp insertion spanning from exon 4 to intron 4 on allele A, resulting in a loss of TK1 activity (Figure 4G). Such mutations cannot be repaired by base editing and are considered challenging even for prime editing. By generating MNs using a combination of allele A-specific sgRNAs (sgLD407E_-4s or sgLD407E_6s), sgS9, and Nickase, we successfully restored TK1 activity in TK6261_LD407E cells. The recovery efficiency of TK1 activity in TK6261_LD407E cells was comparable to that in TK6261 cells (Figure 4G, H). In approximately 80% of the TK1 activity-restored SCCs established after electroporating TK6261_LD407E cells with sgRNAs and Nickase, the long deletion was not detected (Figure 4I). Furthermore, no mutations were found at the breakpoints. These findings demonstrate that MN-IH-HR can potentially repair deletions spanning several hundred base pairs. The data also strongly suggest that MNs can induce MN-IH-HR with long gene conversion tracts.

Minor comments

1. Addition of distances between nicks, mutations, and exons on all the schematics are helpful.

I have added distances between nicks, mutations, and exons to all schemes

2. Schematic for Table 3 is helpful.

We have created a schematic for Table 1 (previously Table 3) in Supplementary Figure

6.

3. *Allele specificities of allele-specific sgRNAs are not fully shown.*

Regarding the verification of sgRNA allele specificity, we have provided detailed information in response to Reviewer 1. In summary, our results are as follows:

- sgEx4_mt20s: allele-specific (Supplementary Figure 1E)
- sgEx5_mt-4s: allele-specific (Supplementary Figure 3C)
- sgEx5_mt28as/TR2: allele-specific (Supplementary Figure 3G)
- sgS43_mt0s: allele-specific (Supplementary Figure 4C)
- sgEx5_WT-4s: allele-specific (Supplementary Figure 5B)
- sgLD407E_-4s: allele-specific (Supplementary Figure 8D)
- sgLD407E_6s: allele-specific (Supplementary Figure 8D)
- sgAP39P_Ex7_mt7s: allele-specific (Supplementary Figure 14A)
- sgFA18JTO_Ex20_mt20s: allele-specific (Supplementary Figure 14B)
- sgEx7_mt121s: allele-specific (This sgRNA is not used in the revised manuscript.)
- sgEx7_WT121s: not allele-specific (This sgRNA is not used in the revised manuscript.)

- sgEx7_WT136as: not allele-specific (This sgRNA is not used in the revised manuscript.)

- sgAT2KYex31_m39as: not allele-specific (This sgRNA is not used in the revised manuscript.)

- Data from the experiments using sgEx7_WT121s, sgEx7_WT136as or sgAT2KYex31_m39as were deleted from the revised manuscript.

4. The last paragraph regarding stem cell in the Discussion section is not relevant to this work at all.

We have removed the last paragraph of the Discussion section.

Reviewer #3 (Remarks to the Author):

The authors present a new gene editing approach that entails the introduction of DNA nicks at the mutated gene allele to trigger gene correction (conversion) directed by a homologous chromosome. This system is suggested to have tangible benefits relative to traditional CRISPR-based gene editing techniques, as mutation correction is characterized by a reduced propensity of secondary mutations as DNA double-strand breaks are, apparently, not induced. The authors demonstrate gene conversion by measuring the recovery of TK1 activity upon introducing one or more targeted DNA nicks. They also find that nick-induced gene conversion has a lesser dependence on canonical homologous recombination factors, such as BRCA1 and BRCA2. Finally, the system was used to correct mutations in known hereditary disease-derived cells.

While nick-induced gene conversion seems like a promising step forward in gene editing, much of what is proposed regarding mechanism is inferred from the data without direct evidence. This is a significant flaw. There is also a possibility that the nicks are converted to one-ended DSBs that are repaired via HR.

Thank you for the constructive comments on our research, and for pointing out the missing data and discussions in our study. We have taken Reviewer 3's important comments seriously and have conducted new experiments.

We used the CDK4/6 inhibitor palbociclib to arrest cells in G1 phase, induced multiple nicks in G1 phase cells, and measured the efficiency of multiple nicks-induced interhomolog recombination (MN-IH-HR). As a result, it was found that the efficiency of MN-IH-HR in G1-arrested cells was about 45% of that in asynchronous cells. Based on these experimental results, we concluded that MN-IH-HR occurs through cell cycle (Figure 5A, B and Supplementary Figure 10, Third paragraph, Page 10).

Knockdown of BRCA1 and BRCA2 reduced IH-HR for paired nick-induced DSBs by 85-90%, while MN-IH-HR was only reduced by 50% (Figure 6A, B). We had interpreted these results as MN-IH-HR being partially dependent on BRCA1 and BRCA2. However, considering the results of the experiments using palbociclib comprehensively, we concluded that at least two pathways are involved in MN-IH-HR: (1) A pathway dependent on the BRCA1 and BRCA2. This pathway is intrinsically the DSB repair pathway. Therefore, the conversion of nick-one-ended DSBs via collision of replication forks should be involved in this pathway. (2) A replication-independent (functioning in G1 phase and probably in G2 phase) pathway that does not depend on BRCA1 and BRCA2.

Based on the newly obtained experimental results and discussion, we have modified the descriptions in the Introduction and Discussion sections. Also, we removed previous proposed model.

Specific Comments

1. The conclusions drawn rely on the premise that multiple nicks are induced and that they are targeted to the correct loci. Can the authors demonstrate this is the case, specifically when they describe targeting one, but not both alleles?

We conducted experiments to confirm that the sgRNA-Cas9 complex accurately recognizes the target sequences for the critical sgRNAs employed in this study. After electroporating wild-type Cas9 and one of the sgRNAs into cells, we analyzed the indel frequency (or the frequency of reads displaying the intact target sequence) using amplicon-based next-generation sequencing (NGS). If the target is precisely recognized by the Cas9-sgRNA complex, the proportion of reads indicating indels will be remarkably high (or the proportion of reads displaying the intact target sequence will decrease significantly). For allele-specific sgRNAs designed to detect only the target sequence on the target allele, the proportion of reads showing the intact target sequence will decrease significantly, while the proportion showing the intact non-target sequence will remain around 50%.

The experiments demonstrated that most examined sgRNAs correctly recognized their target sequences. Any sgRNAs that did not specifically recognize their target sequences were excluded from the study. The data from the amplicon-based NGS are presented in the Figures or Supplementary Figures as below:

- sgEx4_mt20s: Supplementary Figure 1E (allele-specific)
- sgS14: Figure 7B
- sgS34: Figure 7C
- sgS8: Figure 7D
- sgEx5_mt-4s: Supplementary Figure 3C (allele-specific)
- sgEx5_mt28as/TR2: Supplementary Figure 3G (allele-specific)
- sgS43_mt0s: Supplementary Figure 4C (allele-specific)
- sgEx5_WT-4s: Supplementary Figure 5B (allele-specific)
- sgLD407E_-4s: Supplementary Figure 8D (allele-specific)
- sgLD407E_6s: Supplementary Figure 8D (allele-specific)
- sgAP39P_Ex7_mt7s: Supplementary Figure 14A (allele-specific)
- sgFA18JTO_Ex20_mt20s: Supplementary Figure 14B (allele-specific)

2. Conclusions pertaining to the mechanisms and pathways involved need further support. For example, how are the long stretches of ssDNA generated after multiple nicks are presented? EXO1 is involved in both MMR and HR. Does knockdown of EXO1 reduce the MN-IH-HR efficiency (via ssDNA generation), or are there other nucleolytic machineries (e.g. BLM-DNA2) involved in ssDNA generation?

We measured MN-IH-HR efficiency in EXO1 knockout cells and found that EXO1 knockdown decreases MN-IH-HR efficiency (Figure 6C and Supplementary Figure 11C). Our previous manuscript version included data on MN-IH-HR efficiency in BLM knockout cells, showing no effect on MN-IH-HR (Supplementary Figure 11G). We attempted to establish DNA2 knockout cells but were unsuccessful, as homozygous DNA2 knockout has been reported to be embryonic lethal (Lin et al. EMBO J. 15; 32: 1425-1439).

Further investigation is needed to determine if EXO1 is involved in both the BRCA1 and BRCA2-dependent pathway and the BRCA1 and BRCA2-independent pathway. We plan to explore this mechanism in a new research project, as it requires extensive and numerous experiments. However, we note that RPA and RAD51 accumulation between two sgRNA targets (Figure 5C–D, as described later) suggests ssDNA generation via multiple nicking in both homologous chromosomes.

As EXO1-dependent DNA end resection occurs upstream of EXO1-dependent MMR, we cannot determine if EXO1 knockout affects MN-IH-HR efficiency through MMR, similar to MSH2 knockout.

3. Also, if ssDNA stretches are generated, RPA foci should be observable, and they may correlate to the distance between the nicks. Is this true?

We performed CHIP assays using anti-RPA antibodies and found that RPA localized across multiple nicks. However, RPA accumulation was not significant when only one nick was present on a single homologous chromosome (Figure 5C, D, final paragraph, Page 10). This finding suggests that introducing multiple nicks leads to the expansion of exposed ssDNA regions.

We appreciate your suggestion to observe RPA foci formation using fluorescence immunostaining. However, we would like to note that obtaining reliable scientific data through this technique may be challenging for the following reasons:

- Only 1-3 nicks are generated by Nickase within the cells.
- RPA foci can be observed even in cells without exogenously induced DNA damage, making an increase in a few RPA foci statistically insignificant. (For context, in typical experiments

counting RPA foci using fluorescence immunostaining, have experienced significant DNA damage, such as exposure to 3-10 Gy of gamma radiation.)

4. Although the authors do not knockout or inhibit RAD51, they need to compare RAD51 foci after DSB- or nick-induced gene editing.

We conducted a ChIP assay using anti-RAD51 antibodies. Upon inducing a single DSB in the genome, RAD51 accumulation at the DSB site was barely observed (Figure 5E, right panel). Since most DSBs generated by Cas9 are repaired with mutations through NHEJ, as demonstrated in Figure 7, the RAD51-dependent HR pathway is infrequently utilized for Cas9-induced DSB repair. Moreover, when a single nick was introduced into the genome, RAD51 accumulation at the target site was scarcely detected. Conversely, when multiple nicks were induced on both homologous chromosomes, RAD51 accumulation was observed at the sgRNA target sites and the region between the targets. These findings suggest that MN-IH-HR involves a RAD51-dependent DNA repair pathway, and multiple nicks are more frequently repaired via the RAD51-dependent repair pathway than DSBs.

It is important to note that there is variability in the results of the ChIP assay-digital PCR, as shown in Figures 5D and E. It is suggested that using 10 times the amount of cells and antibody may yield more consistent results. However, it should be noted that conducting such experiments is not practical due to the cost of consumables, which can amount to several thousand dollars per sample.

5. Supplementary Figure 13 A-C show that CtIP-AID, and BRCA1-AID cells have significantly less CtIP and BRCA1 expression compared to TSCER2 cells, even without auxin treatment, indicating these are leaky degrons.

As Reviewer 3 has pointed out, it is conceivable that BRCA1-AID, BRCA2-AID, and CtIP-AID might be degraded due to leaky degrons. Nevertheless, the reduced expression levels of BRCA1-AID, BRCA2-AID, and CtIP-AID are sufficient to support interhomolog HR (IH-HR) of paired nick-induced DSBs (pnDSBs) (Figure 6A, B). Moreover, further degradation of BRCA1-AID, BRCA2-AID, and CtIP-AID by auxin degrons can almost completely inhibit

IH-HR of pnDSBs (Figure 6A, B). Consequently, it is deemed appropriate to use these cells to validate the BRCA1, BRCA2, and CtIP dependency of MN-IH-HR in scientific research.

6. In the Discussion, the author state that “MNs are repaired in absence of BRCA2”, however, the repair was significantly reduced with BRCA2 depletion as shown in 6D. Please clarify.

As shown in Figure 6A B, the decrease in MN-IH-HR efficiency due to BRCA1 or BRCA2 knockdown is approximately 50% (compared to 85-90% reduction for pnDSBs). This cannot be explained by a single mechanism. We propose the existence of at least two MN-IH-HR pathways (Supplementary Figure 15A):

1. A pathway reliant on the BRCA1-BRCA2-RAD51 axis. The BRCA1 and BRCA2-dependent HR pathway is inherently the DSB repair pathway. Thus, the conversion of nick-one-ended DSBs through replication fork collisions should be involved in this pathway. This process occurs exclusively in the S phase. Indeed, when cells were arrested in the G1 phase, MN-IH-HR efficiency was reduced.

2. A pathway independent of BRCA1 and BRCA2. Multiple nicks stimulate MN-IH-HR in G1 arrested cells, although this is less efficient than in asynchronous cells (Figure 5A, B). In the G1 phase, nicks are not converted to one-ended DSBs via replication. Previous studies have identified the presence of BRCA2 and RAD51-independent nick-induced HDR (Davis et al. Cell Rep 2016). Therefore, an alternative pathway independent of BRCA1 and BRCA2 is considered to involve HR-like interhomolog recombination without nick-to-DSB conversion.

Please refer to the Discussion section for a comprehensive and diverse examination of the mechanisms based on the obtained data.

7. Depletion of XRCC1 and PARP1 increases MN-IH-HR suggesting SSBR pathway completes with this mechanism, as shown in Supplemental Fig 13K. Since the authors have not presented any data demonstrating that MNs are really generated at the said loci, it would be helpful to confirm the same via XRCC1 and PARP1 ChIP studies.

We performed ChIP assays using five types of PARP1 antibodies; however, the high background levels rendered data analysis impossible. Similarly, we could not find a suitable XRCC1 antibody for ChIP assay. Nonetheless, ChIP assays for RPA revealed an expansion of ssDNA regions upon the induction of MNs, suggesting that MNs were introduced as intended.

As previously described, we verified that sgEx4_mt20s and sgS14 detect and cleave DNA at their target sites precisely with Cas9 (Supplementary Figure 1E, 7B).

Reviewer #3 suggested demonstrating the occurrence of multiple nicks at the target sequences by ChIP assays using antibodies that detect nicks. However, in ChIP assays using these antibodies and cell populations, we cannot distinguish between the following two cases: nicks occurring simultaneously at targets A and B in a portion of cell populations; nicks occurring only at target A in a subset of cell populations and only at target B in another subset of cell populations. As a result, it is logically impossible to prove the simultaneous occurrence of nicks within cells using ChIP assays.

We have excluded the data from PARP1^{-/-}PARP2^{+/+} cells from our study, considering the feedback from Reviewer 1. Additionally, we have removed the corresponding model figure.

8. The nicks could be converted to one-ended DSB in dividing cells that would be repaired via HR. Repeated introduction on nicks increases the gene conversion event, which could be via increasing the possibility of generation of one-ended DSB, instead of repair via SSBR pathway. Furthermore, reduction of gene conversion events when the nicks were introduced in the other allele, as shown in Figure 5, may also indicate this to be true. Discuss this point at the minimum.

As Reviewer #3 suggested, We believe that that the BRCA1 and BRCA2-dependent pathway involves the conversion of nicks to one-ended DSBs through replication fork collision. Consequently, we have included a discussion in the manuscript to address this possibility (Page 16-17).

Minor Comments

1. The title is cumbersome and not a simple representation of the work. Suggestion: "Inducing multiple nicks promotes interhomolog recombination to correct heterozygous mutations without DNA double-strand breaks."

Thank you for suggesting a better title. We have decided to adopt this title with minor modifications.

2. Labelling of Supplementary Figure 13 F and G are swapped.

Thank you for pointing out the error. We have made the appropriate corrections in the new manuscript.

Reviewers' Comments:

Reviewer #1:

Remarks to the Author:

Tomita et al. have substantially improved their manuscript by quality controlling their experimental settings and strengthening their findings. Regarding the former aspect, I highlight the fact that they have formally established whether the initially presumed allelic specificity of the primary gRNAs is factual (or not), by using the proposed quantification of Cas9 nuclease-derived indels as a readout. These QC experiments allowed them purging datasets obtained with two gRNAs that turned out to lack allelic specificity.

Concerning the strengthening of their findings, I highlight the implementation of ChIP-based assays to present a direct correlation between multiple genomic nicks and the accumulation of RPA and RAD51 proteins at, and in-between, target sequences.

Presently, I only have a couple of comments for further improvement, namely, in the Discussion section and take the opportunity to indicate specific small edits in the main text. These are detailed next.

The authors put forward two models underlying MN-IH-HR-based genome editing. One of these working models involves BRCA1- and BRCA2-dependent repair of one-ended DSBs generated when replication forks hit induced nicks and collapse. Such events are expected to be most problematic at off-target nicking sites. Although the MN-IH-HR approach offers clear advantages over canonical DSB-based genome editing, the reliance on multiple Cas9.D10A:gRNA complexes naturally increases the risk for off-target one-ended DSB formation. Significantly, this risk might be substantially reduced by using high-specificity Cas9.D10A nickases instead of the regular, more off-target-prone, Cas9.D10A nickase (PMID: 33398349). Moreover, such high-specificity nickases should also confer enhanced target-allele specificity to primary gRNAs owing to their heightened discrimination of spacer-protospacer mismatches. Hence, a brief discussion on MN-IH-HR limitations/caveats (i.e., gRNA-dependent generation of one-ended off-target DSBs and reduced target allele selectivity caused by poor discrimination between alleles with single or few nucleotide variations) followed by a potentially straightforward solution involving the use of high-specificity nickases (PMID: 33398349), will contribute to an improved and more balanced Discussion.

Also to better frame the author's findings in relation to earlier research consistent with these findings, the text that reads:

"The findings reveal that SNs on both homologous chromosomes promote MN-IH-HR, but tandem nicks on one chromosome do not (Figure 3D-G), suggesting that the secondary nick on the allele without the primary nick is crucial for MN-IH-HR enhancement. Although the detailed mechanisms remain unclear, exposure of ssDNA surrounding the MNs on both homologous chromosomes (Figure 5C-E) likely contribute to increasing the occurrence of MN-IH-HR."

Will profit by introducing a brief sentence(s) (before or after) referring to research showing that targeted nicking of endogenous genomic DNA induces homology-directed recombination (revised version Ref. 18/PMID: 19651880) and that simultaneous nicking of recipient and exogenous donor DNA templates fosters this process (revised version Ref. 22/PMID: 22189101).

Edits:

Page 6, line 132: add bp after 8,675.

Page 7, line 158: perhaps better to substitute "times" by "fold".

Page 17, line 431: perhaps better to substitute "muscular" by "muscle" and associate the indicated cells with the words "non-dividing" or "post-mitotic" to make the argument dispensed in this sentence more explicit.

Finally, I would like to congratulate Tomita and colleagues for their thorough and important work.

Manuel A.F.V. Gonçalves (reviewer 1)

Reviewer #2:

Remarks to the Author:

I sincerely appreciate the author's efforts on this revision to address reviewer's criticisms with numerous new data and would like to celebrate their successful revision. I'm very satisfied by the authors responses and especially love the data to repair large deletion by MN-IH-HR (Fig4G-I). However, the responses to my major comment #1 still don't address my questions. Before publication, I would like to ask a minor revision on these comments.

Also, I suggest the authors and the editor to give an attractive and catchy nickname to this technology, rather than just MN-IH-HR. There are so many good examples in this field (for example, base and prime editing by David Liu, CAST, SHERLOCK, REPAIR and SEND by Feng Zhang, PASTE and RADAR by Omar Abudayyeh and Jonathan Gootenberg, INTEGRATE by Sam Sternberg etc)

Major comment #1:

•Regarding "the old Supplementary Figure 13 (Supplementary Figure 15B in the revised manuscript)":

The most important DNA repair mechanism which the readers want to know is how multiple nicks, not single nick, improve IH-HR. However, the schematics the authors propose still do not directly answer this question. In the new Supplementary Figure 15A, the authors propose nick-to-DSB conversion as a mechanism of MN-IH-HR, however, this should also happen in the case of single nick. This is same for the case without nick-to-DSB conversion. In the new Supplementary Figure 15B, the authors summarize how MMR resolves the mismatch after MN-IH-HR, however, the field already knows this (for example, see Britt Adamson and David Liu's Repair-seq work (PMID: 34653350)). The authors should provide a model directly explaining how multiple nicks improve IH-HR. As the authors propose the expanded ssDNA exposure induces more RPA binding, which is eventually replaced with RAD51, is the key mechanism of MN-enhanced IH-HR, they should incorporate this into their model. It's also interesting to test whether RAD51 overexpression stimulates MN-IH-HR as reported in the case of DSB-mediated IH-HR (in zygotes, ref. 46).

Why SNs on both alleles also increases RPA and RAD51 binding? This should not change the length of ssDNA exposure.

In the author's response, Figure 5B should be Figures 5D and 5E?

•Regarding "Cas9H840A":

The new data nicking antisense strand is very exciting and consistent with MMR theory (Fig 5, and previous base editing and prime editing works). But, in this case, how IH-HR can be enhanced? Multiple nicks on opposite strand do not expand ssDNA exposure followed by RPA binding. Does this strategy enhance IH-HR by biasing MMR which was done in base and prime editing? I respect the author's decision not including this into the current manuscript, however, they should be aware that Nature Communications may publish reviewer's comments and the author's responses.

In this figure, the sgRNA name Ex4_20s is correct? Not Ex4_mt20s?

Reviewer #3:

Remarks to the Author:

The authors have done a large amount of work to address the main points of this reviewer. The paper is now suited to publication.

Point-by-point response

Tomita et al. Inducing multiple nicks promotes interhomolog homologous recombination to correct heterozygous mutations in somatic cells

We appreciate the positive feedback from all reviewers regarding our revised manuscript. Below, we offer point-by-point responses to each comment from the reviewers. In response to Reviewer #2's suggestion, we reconsidered the name of the MN-IH-HR method for correcting heterozygous mutations, and have decided to rename it as NICER (an approach for correcting heterozygous mutations, by employing multiple nicks induced by Cas9 nickase and the homologous chromosome as an endogenous repair template). In order to address the concerns raised by the reviewer, we have expanded the discussion and made revisions to the model schematic. In compliance with the standards set by the Human Genome Variation Society (HGVS), we have updated the notation of the genetic mutation; specifically, 'c.231_232insC' is now denoted as 'c.231dupC', 'c.311_318del8' has been changed to 'c.311_318del', and 'c.640_641insG' has been revised to 'c.640dupG'.

The original comments from the reviewers are presented in green, while the authors' responses are shown in black. We are grateful for the opportunity to have our manuscript reconsidered for publication in Nature Communications.

Reviewer #1 (Remarks to the Author):

Tomita et al. have substantially improved their manuscript by quality controlling their experimental settings and strengthening their findings. Regarding the former aspect, I highlight the fact that they have formally established whether the initially presumed allelic specificity of the primary gRNAs is factual (or not), by using the proposed quantification of Cas9 nuclease-derived indels as a readout. These QC experiments allowed them purging datasets obtained with two gRNAs that turned out to lack allelic specificity.

Concerning the strengthening of their findings, I highlight the implementation of ChIP-based assays to present a direct correlation between multiple genomic nicks and the accumulation of RPA and RAD51 proteins at, and in-between, target sequences.

Presently, I only have a couple of comments for further improvement, namely, in the Discussion section and take the opportunity to indicate specific small edits in the main text. These are detailed next.

The authors put forward two models underlying MN-IH-HR-based genome editing. One of these working models involves BRCA1- and BRCA2-dependent repair of one-ended DSBs generated when replication forks hit induced nicks and collapse. Such events are expected to be most problematic at off-target nicking sites. Although the MN-IH-HR approach offers clear advantages over canonical DSB-based genome editing, the reliance on multiple Cas9.D10A:gRNA complexes naturally increases the risk for off-target one-ended DSB formation. Significantly, this risk might be substantially reduced by using high-specificity Cas9.D10A nickases instead of the regular, more off-target-prone,

Cas9.D10A nickase (PMID: 33398349). Moreover, such high-specificity nickases should also confer enhanced target-allele specificity to primary gRNAs owing to their heightened discrimination of spacer-protospacer mismatches. Hence, a brief discussion on MN-IH-HR limitations/caveats (i.e., gRNA-dependent generation of one-ended off-target DSBs and reduced target allele selectivity caused by poor discrimination between alleles with single or few nucleotide variations) followed by a potentially straightforward solution involving the use of high-specificity nickases (PMID: 33398349), will contribute to an improved and more balanced Discussion.

We are sincerely grateful for the insightful comments provided by Reviewer #1. We fully concur with the points raised and appreciate the guidance given. Indeed, several months ago, we embarked on our research utilizing high-fidelity nickases. Responding to Reviewer #1's valuable suggestion, we have incorporated an expanded discussion with pertinent references into the third paragraph of the Discussion section.

Also to better frame the author's findings in relation to earlier research consistent with these findings, the text that reads:

“The findings reveal that SNs on both homologous chromosomes promote MN-IH-HR, but tandem nicks on one chromosome do not (Figure 3D–G), suggesting that the secondary nick on the allele without the primary nick is crucial for MN-IH-HR enhancement. Although the detailed mechanisms remain unclear, exposure of ssDNA surrounding the MNs on both homologous chromosomes (Figure 5C–E) likely contribute to increasing the occurrence of MN-IH-HR.”

Will profit by introducing a brief sentence(s) (before or after) referring to research showing that targeted nicking of endogenous genomic DNA induces homology-directed recombination (revised version Ref. 18/PMID: 19651880) and that simultaneous nicking of recipient and exogenous donor DNA templates fosters this process (revised version Ref. 22/PMID: 22189101).

We have added a brief statement to the Discussion section, citing relevant literatures (including PMID: 19651880 and 22189101) that aligns with the findings presented in this paper.

Edits:

Page 6, line 132: add bp after 8,675.

Page 7, line 158: perhaps better to substitute “times” by “fold”.

Page 17, line 431: perhaps better to substitute “muscular” by “muscle” and associate the indicated cells with the words “non-dividing” or “post-mitotic” to make the argument dispensed in this sentence more explicit.

Thank you for your feedback. We have made the necessary revisions to the manuscript based on your suggestions.

*Finally, I would like to congratulate Tomita and colleagues for their thorough and important work.
Manuel A.F.V. Gonçalves (reviewer 1)*

Reviewer 1's comment was constructive, paying attention to detail, and extremely valuable, making a significant contribution to greatly improving our research. We, the authors, are truly grateful to Reviewer 1.

Reviewer #2 (Remarks to the Author):

I sincerely appreciate the author's efforts on this revision to address reviewer's criticisms with numerous new data and would like to celebrate their successful revision. I'm very satisfied by the authors responses and especially love the data to repair large deletion by MN-IH-HR (Fig4G-I). However, the responses to my major comment #1 still don't address my questions. Before publication, I would like to ask a minor revision on these comments.

Also, I suggest the authors and the editor to give an attractive and catchy nickname to this technology, rather than just MN-IH-HR. There are so many good examples in this field (for example, base and prime editing by David Liu, CAST, SHERLOCK, REPAIR and SEND by Feng Zhang, PASTE and RADAR by Omar Abudayyeh and Jonathan Gootenberg, INTEGRATE by Sam Sternberg etc)

Major comment #1:

•Regarding “the old Supplementary Figure 13 (Supplementary Figure 15B in the revised manuscript)”:

The most important DNA repair mechanism which the readers want to know is how multiple nicks, not single nick, improve IH-HR. However, the schematics the authors propose still do not directly answer this question. In the new Supplementary Figure 15A, the authors propose nick-to-DSB conversion as a mechanism of MN-IH-HR, however, this should also happen in the case of single nick. This is same for the case without nick-to-DSB conversion. In the new Supplementary Figure 15B, the authors summarize how MMR resolves the mismatch after MN-IH-HR, however, the field already knows this (for example, see Britt Adamson and David Liu's Repair-seq work (PMID: 34653350)). The authors should provide a model directly explaining how multiple nicks improve IH-HR. As the authors propose the expanded ssDNA exposure induces more RPA binding, which is eventually replaced with RAD51, is the key mechanism of MN-enhanced IH-HR, they should incorporate this into their model. It's also interesting to test whether RAD51 overexpression stimulates MN-IH-HR as reported in the case of DSB-mediated IH-HR (in zygotes, ref. 46).

We thank Reviewer #2 for positive responses to our revised manuscript

We have named the gene editing method utilizing MN-IH-HR as NICER (an approach for correcting heterozygous mutations, by employing multiple nicks induced by Cas9 nickase and the homologous chromosome as an endogenous repair template).

In our previous manuscript, we regret that we did not thoroughly express our insights on the notable increase in MN-IH-HR frequency when multiple nicks are introduced to homologous chromosomes. Furthermore, we failed to provide an explanation for the scenario where a single nick was introduced solely on the mutant allele.

In our most recent manuscript, we have rectified this by providing a detailed explanation. When multiple nicks are introduced on both homologous chromosomes, the alleles, whether acting as the recipient or the donor, expose long ssDNA strands. Consequently, we anticipate an increase in the opportunities for the ssDNA strand on the recipient allele to anneal to the ssDNA strand on the donor allele. However, when a single nick is introduced exclusively on the mutant allele, the supposed donor allele does not expose ssDNA strands, suggesting that such annealing opportunities likely will not increase. This explanation is elaborated in the final paragraph on page 16 of our revised manuscript.

To help our readers understand this concept more visually and intuitively, we have updated the schematic in Supplementary Figure 15A, where we have now included RPA and RAD51.

The idea of stimulating MN-IH-HR through the overexpression of RAD51 is indeed intriguing. However, our unpublished data reveals that inducing RAD51 overexpression in TK6 cells results in cell death, preventing us from confirming this approach.

We hope these clarifications and revisions provide a more comprehensive understanding of our NICER gene editing method, which uses MN-IH-HR to correct heterozygous mutations.

Why SNs on both alleles also increases RPA and RAD51 binding? This should not change the length of ssDNA exposure.

Firstly, we would like to convey that the exposure of an ssDNA strand does not necessarily imply the loss of one of the strands from the double-stranded DNA. During the unwinding of the DNA double helix, both DNA strands become exposed as ssDNA strands, a process that occurs without any disappearance or removal of either DNA strand.

Our ChIP-digital PCR results (Figure 5 C–E) show a greater binding of RPA and RAD51 to the target region when multiple nicks are introduced to both chromosomes, as well as when single nicks are introduced to both chromosomes. We interpret these findings to

suggest that long ssDNA exposure occurs on both homologous chromosomes when single nicks are introduced to both. We propose that the exposure of long ssDNA strand can also occur when a single nick is introduced by nickase (but ssDNA strands are not exposed on the homologous chromosome without a nick). This theory is also supported by Davis et al. (PNAS 2014 [PMID: 24556991] and Cell Rep 2016 [PMID: 27829157]). When nicks are introduced into both chromosomes, the exposure of ssDNA strands potentially doubles compared to when a single nick is introduced only to one chromosome. We believe our ChIP-digital PCR assay has detected this difference. (Please note that the amount of DNA detected by the ChIP-digital PCR assay may not directly correlate with the amount of exposed ssDNA.)

However, one might wonder why efficient gene editing isn't achieved with the combination of sgS14 and nickase. Even if ssDNA strands are exposed in the region surrounding the c.231dupC (c.231_232insC) mutation when both alleles are singly nicked by an sgS14-nickase complex, there isn't a nick or DSB near the c.231dupC (c.231_232insC) mutation site. Therefore, the c.231dupC (c.231_232insC) mutation, which is located more than 1,600 bp downstream of the sgS14 target site, will not be overwritten with the wild-type sequence through IH-HR. Consequently, we surmise that TK1 activity isn't efficiently restored by introducing single nicks at the sgS14 target sites (Figure 2D). We present a schematic of our proposed model here. Given the complexity already inherent in our model detailed in Supplementary Figure 15A, we have chosen not to incorporate the model shown below into Supplementary Figure 15A.

In the author's response, Figure 5B should be Figures 5D and 5E?

As Reviewer #2 pointed out, it was Figure 5D and 5E, not Figure 5B. I apologize for the mistake.

•Regarding “Cas9H840A”:

The new data nicking antisense strand is very exciting and consistent with MMR theory (Fig 5, and previous base editing and prime editing works). But, in this case, how IH-HR can be enhanced? Multiple nicks on opposite strand do not expand ssDNA exposure followed by RPA binding. Does this strategy enhance IH-HR by biasing MMR which was done in base and prime editing?

I respect the author’s decision not including this into the current manuscript, however, they should be aware that Nature Communications may publish reviewer’s comments and the author’s responses.

In this figure, the sgRNA name Ex4_20s is correct? Not Ex4_mt20s?

Thank you for your comments. Since they contain unpublished data, I am unable to provide detailed information here. Nevertheless, we have confirmed

[Redacted]

As Reviewer #2 pointed out, “sgEx4_20s” should be “sgEx4_mt20s”. I apologize for the mistake.

(Due to the inclusion of confidential data and ideas in the response to this question, I kindly request that they be redacted when published in the peer review file.)

Reviewer #3 (Remarks to the Author):

The authors have done a large amount of work to address the main points of this reviewer. The paper is now suited to publication.

Thank you for all valuable suggestions.

Reviewers' Comments:

Reviewer #2:

Remarks to the Author:

All my questions are appropriately resolved now. Thank you very much for giving me this opportunity to enjoy the nice NICER manuscript. Once again, I would like to celebrate the authors for their successful revision and great works.

Just a comment on their response regarding "one might wonder why efficient gene editing isn't achieved with the combination of sgS14 and nickase." The author's response "Even if ssDNA strands are exposed in the region surrounding the c.232dupC (c.231_232insC) mutation when both alleles are singly nicked by an sgS14-nickase complex, there isn't a nick or DSB near the c.232dupC (c.231_232insC) mutation site. Therefore, the c.232dupC (c.231_232insC) mutation, which is located more than 1,600 bp downstream of the sgS14 target site, will not be overwritten with the wild-type sequence through IH-HR." is probably correct. RPA binding in PCR I in Fig5D supports this idea (less RPA binding compared to the one by MNs and comparable RPA binding to the one by SN on allele A). However, this data is not incorporated in their model scheme, and RPA binding is described spanning across PCR I-III, making me confused.

Point-by-point response

Tomita et al. Inducing multiple nicks promotes interhomolog homologous recombination to correct heterozygous mutations in somatic cells

We appreciate the feedback from Reviewer #2 on our revised manuscript. Below, we provide a point-by-point response to the comment raised by the reviewer. The original comment from the reviewer is presented in green text, while our response is in black.

We are grateful for the chance to have our manuscript re-evaluated for publication in Nature Communications.

Reviewer #2 (Remarks to the Author):

All my questions are appropriately resolved now. Thank you very much for giving me this opportunity to enjoy the nice NICER manuscript. Once again, I would like to celebrate the authors for their successful revision and great works.

Just a comment on their response regarding “one might wonder why efficient gene editing isn't achieved with the combination of sgS14 and nickase.” The author's response “Even if ssDNA strands are exposed in the region surrounding the c.232dupC (c.231_232insC) mutation when both alleles are singly nicked by an sgS14-nickase complex, there isn't a nick or DSB near the c.232dupC (c.231_232insC) mutation site. Therefore, the c.232dupC (c.231_232insC) mutation, which is located more than 1,600 bp downstream of the sgS14 target site, will not be overwritten with the wild-type sequence through IH-HR.” is probably correct. RPA binding in PCR I in Fig5D supports this idea (less RPA binding compared to the one by MNs and comparable RPA binding to the one by SN on allele A). However, this data is not incorporated in their model scheme, and RPA binding is described spanning across PCR I-III, making me confused.

We sincerely appreciate your positive evaluation of our paper and the insightful comments provided by Reviewer #2. We revised the model scheme in Supplementary Figure 17. We also have incorporated the following description, addressing the reviewer's concern:

"When the sgS14-Nickase complex was employed to introduce SNs on both alleles, RPA accumulation was observed in regions I and II (between the target sites of sgS14 and sgEx4_mt20s, Figure 5c) but not in region I (near the target site of sgEx4_mt20s, Figure 5c) (Figure 5d). When MNs were generated by sgEx4_mt20s, sgS14, and Nickase, RPA accumulation was detected not only at the nicked sites (regions I and III) but also in the regions between the two target sites (region II) (Figure 5d)."

We believe that this revised description adequately addresses the concern raised by Reviewer

#2.